# Multiscale statistical analysis of thermal and non-thermal components of seawater $p\mathrm{CO}_2$ in the Western English Channel: scaling, time-reversibility, and dependence

Kévin Robache and François G. Schmitt

Laboratoire d'Océanologie et Géosciences, Université du Littoral Côte d'Opale, Université de Lille, CNRS, IRD, UMR LOG 8187, F62930 Wimereux, France

**Correspondence:** Kévin Robache (kevin.robache@univ-littoral.fr) and François G. Schmitt (francois.schmitt@cnrs.fr)

**Abstract.** High-frequency variability of the partial pressure of $\mathrm{CO}_2$ ($p\mathrm{CO}_2$) in coastal environments reflects the complex interplay of physical, chemical and biological drivers. Multiscale statistical approaches provide a robust framework for understanding dynamics across timescales and for reliably assessing coastal carbon processes. In this study, $p\mathrm{CO}_2$ has been measured on the Astan cardinal buoy (Brittany, west coast of France) with at 30-minute intervals by Gac et al. (2020), yielding a dataset of 32 582 data points collected over a period of nearly five years. These measurements were then coupled with others of sea surface temperature and salinity, chlorophyll $a$, oxygen saturation and atmospheric pressure. The aim of this study was to consider the statistical properties of the thermal and non-thermal component of $p\mathrm{CO}_2$, based on its relation with temperature established by Takahashi et al. (2009). Using Fourier spectral analysis, it was demonstrated that all marine scalars exhibited scaling properties with power-law slopes ranging from 1.73 to 1.85 for timescales spanning from 12 hours to at least 80-100 days. The results obtained from this analysis indicate a turbulent and intermittent dynamics for all the considered scalars, including sea surface temperature and salinity, chlorophyll $a$, oxygen saturation, $p\mathrm{CO}_2$, and $p\mathrm{CO}_2$ thermal and non-thermal components. A time-reversibility analysis evidenced the irreversibility of the $p\mathrm{CO}_2$ components above 30 days. The irreversibility exhibited by the thermal component was found to be higher than that of the non-thermal component, with an average value of the associated irreversibility index that was approximately 3.5 times higher than that of the non-thermal component over the period of 50 to 70 days. Furthermore, a methodology known as the Probability Density Function quotient was employed, a method that has not been widely utilized. This approach enabled the identification of values for which there were statistical relationships between variables. This facilitated the quantification of the influence of primary production on the non-thermal $p\mathrm{CO}_2$, or the influence of periods of depression on supersaturation due to atmospheric or terrigenous inputs. This provided new insights into the stochastic coupling between biological and physical processes, when considering high-frequency $p\mathrm{CO}_2$ variability.

## 1 Introduction

The ocean carbon cycle plays a major role in the global carbon cycle of the Earth. It encompasses a wide array of physico-chemical, biological, and geological processes, each operating at various spatio-temporal scales ranging from the largest to the finest. Examples include extreme events (Yu et al., 2020) and phytoplankton blooms (Bozec et al., 2011). Its significance is

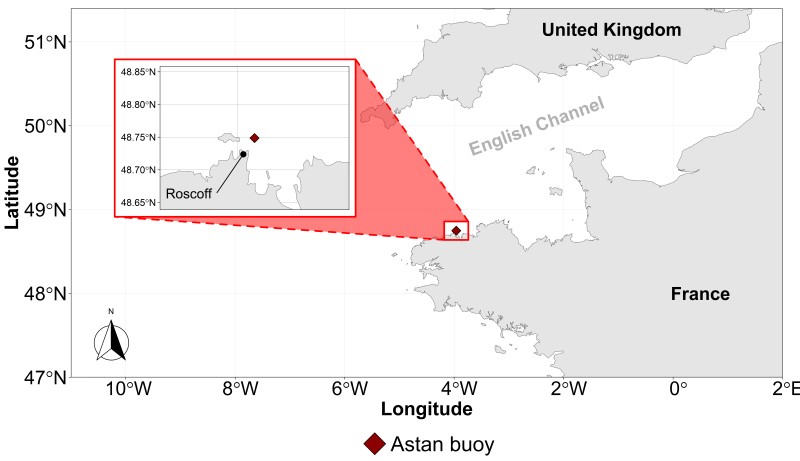

**Figure 1.** A map of western France indicating the location of the Astan buoy off the coast of the city of Roscoff.

crucial for global climate regulation, particularly in light of the pressing issue of global warming. The ocean's role is paramount
as it serves as a carbon sink, capable of absorbing approximately 25 % of atmospheric anthropogenic carbon emissions each
year, according to recent estimations (Friedlingstein et al., 2023). To anticipate future dynamics of carbon dioxide ($CO_2$),
models serve as valuable tools, such as those used in the Shared Socio-economic Pathways (SSP) scenarios of the Coupled
Model Intercomparison Project Phase 6 (CMIP6) used in the Intergovernmental Panel on Climate Change reports (IPCC, 2023).
However, constructing and/or improving these models requires *in situ* observations (Ciais et al., 2014). Time series analysis,
in particular, is needed to assess the multiscale dynamics of monitored quantities. High-frequency time series are therefore
essential for constructing reliable models with precise spatial and temporal resolutions. For several years now, efforts are made
in this direction to build large observation databases such as the Surface Ocean $CO_2$ Atlas (SOCAT; Bakker et al., 2016)
database, using in part autonomous moorings (Sutton et al., 2014, 2019; Gac et al., 2020). Furthermore, it is really important
to record Eulerian (e.g., fixed buoy) or Lagrangian (e.g., drifting buoy) time series to clearly understand the temporal behavior
of the marine scalars.

Due to their large variability at all scales, coastal areas are crucial regions to monitor in order to obtain the most accurate
carbon balance. These areas are subject to various events having an impact on $CO_2$ dynamics both in time and space. For
example, anthropogenic activities (e.g., industry and agriculture; Northcott et al., 2019), land inputs (Jiang et al., 2013), extreme
events (e.g., storms; Crosswell et al., 2014) in these areas are factors of fast $CO_2$ fluctuations. Furthermore, these zones possess
a high seasonality due to different seasonal drivers such as biological activity and sea surface temperature across seasons
(Bozec et al., 2011; Marrec et al., 2013; Reimer et al., 2017; Torres et al., 2021). Moreover, these areas can also be particularly
turbulent. This can impact the variability across different timescales and the properties of the scalars (e.g., reversibility, scaling
properties and intermittency; Zongo and Schmitt, 2011; Schmitt and Huang, 2016; Schmitt, 2023) such as seawater $CO_2$
(Robache et al., 2025). Hence these zones present a complex multiscale variability which needs to be described and understood
as far as possible.

Several studies have shown that thermal variations contribute to fluctuations in the partial pressure of $CO_2$ ($pCO_2$) at the seawater surface (Takahashi et al., 1993, 2002, 2009; Wanninkhof et al., 2022). However, as previously mentioned, numerous other physical, chemical, biological, and geological events which are not directly linked to temperature dynamics influence the dynamics of $pCO_2$. These include processes such as photosynthesis and respiration, as well as air-sea $CO_2$ exchanges

(Gac et al., 2020). The variations in the seawater partial pressure of $pCO_2$ can thus be classified into two groups: thermal and non-thermal components. In this study, we aim to investigate the dynamics of these two components, and to compare them, using statistical tools from the field of turbulent time series analysis. For this, a coastal database situated in the western English Channel (France) is considered, in order to study an example in a high-variability environment.

The structure of the paper is the following: Sect. 2 presents the data and methods employed in this study, with a detailed

description of the approach used to quantify the non-thermal variations of $pCO_2$ provided in Sect. 2.2. The statistical analysis of these two components is presented in Sect. 3, with a focus on multivariate analysis in Sect. 3.2. Finally, the last sections provide the discussion (Sect. 4) and conclusions (Sect. 5).

## 2   Data and Methods

### 2.1   Presentation of the database from the Astan cardinal buoy

The data analyzed here belong to the Astan cardinal buoy, which is situated off the coast of Brittany, France, within the Western English Channel (WEC). Specifically, it is positioned 3.1 km offshore Roscoff (48°44'55" N and 3°57'40" W) as depicted in Fig. 1. The WEC is characterized as a turbulent shallow epicontinental sea (Dauvin, 2019). The average depth at the buoy's location is 45 m, and this region experiences pronounced tidal influences, with tidal ranges exceeding 8 m during spring tides (Gac et al., 2020). Consequently, an intense tidal stream is consistently present, facilitating the continuous

total vertical mixing of the water column (Pingree and Griffiths, 1978; Reid et al., 1993; Marrec et al., 2013). This buoy is an integral component of the French national observation service COAST-HF, within the IR-ILICO coastal and marine research infrastructure. The seawater surface partial pressure of $CO_2$ ($pCO_2$) data were collected on the buoy using a SAMI-$CO_2$ (Submersible Autonomous Moored Instrument, SunBurst Sensors) sensor, recording measurements every 30 minutes at 6 m depth. This device monitors $pCO_2$ by detecting pH changes (via absorbance measures) in a bromothymol blue solution

contained within a gas-permeable membrane, induced by the presence of $CO_2$. Sea Surface Temperature (SST) and Sea Surface Salinity (SSS) were simultaneously measured using a SeaBird Inc. SBE16+ sensor (at 5 m depth), at the same time resolution, and Dissolved Oxygen (DO) and chlorophyll $a$ (chl $a$, based on fluorescence and calibrated using low-frequency chlorophyll data; see Gac et al., 2020) concentrations were recorded with a SeaBird Inc. SBE43 sensor and a Cyclops7 sensor (both at 5 m depth), respectively (Gac et al., 2020). To mitigate potential issues related to ambient light, a cover was placed over the

fluorescence sensor and the measurements were carried out in an opaque chamber. Oxygen saturation ($O_{sat}$) was estimated by Gac et al. (2020) following Weiss (1970). Atmospheric pressure ($P_{atm}$) data were provided by the weather station of the Roscoff Marine Station (at 15 m height).

**Table 1.** Number of recorded data points ($n$) and percentage of missing data ($\%_{\text{missing}}$) for the time series used for the following of this study.

| Scalar | $n$ | $\%_{\text{missing}}$ |
|---|---|---|
| $p\text{CO}_2$ | 32 582 | 61.5 |
| Chl $a$ | 37 412 | 55.8 |
| SST | 62 981 | 25.5 |
| SSS | 62 475 | 26.1 |
| $\text{O}_{\text{sat}}$ | 50 752 | 40.0 |
| $\text{P}_{\text{atm}}$ | 68 463 | 19.0 |

This database was established by Gac et al. (2020); they assessed the dataset's reliability through a comparison with low-frequency (e.g., weekly) sampling observations with laboratory estimations of $p\text{CO}_2$. Moreover, blank measurements were regularly performed in distilled water for quality control, with sensors retrieved every three months for inspection, cleaning, battery checks, and reagent level verification. More technical details can be found in Gac et al. (2020). The considered database is thus originally composed of six marine and atmospheric parameters ($p\text{CO}_2$, SST, SSS, $\text{O}_{\text{sat}}$, chl $a$, $\text{P}_{\text{atm}}$) recorded from March 6, 2015 to December 31, 2019 with a time step of 30 minutes. If there were no missing data, the total number of data points would be 84 552. However, due to local failures of the measuring devices, there are missing data for each parameter. Table 1 presents for each parameter the total number of recorded data points, and the associated percentage of missing values: this percentage varies from 19.0 % for $\text{P}_{\text{atm}}$ to 61.5 % for $p\text{CO}_2$. Due to this large proportion of missing values we are not considering here procedures to fill the gaps. The data analysis methods must then be adapted to time series with large portions of missing values. In addition, the intervals between missing values vary considerably in length, with some being long and others short. As a result, the influence of missing values is distributed across all scales. Overall, despite approximately 60 % of the $p\text{CO}_2$ data being missing, the statistics at each scale (computed from the increments between available values) are still based on a sufficiently large number of observations. Further discussion of this issue, including details on the number of data points at each scale, is provided in Appendix A.

## 2.2 A non-thermal $p\text{CO}_2$ index

We denote $p\text{CO}_2^{\text{thermal}}$ the changes of $p\text{CO}_2$ induced by temperature changes, under isochemical conditions. This is written as an exponential relation:

$$p\text{CO}_2^{\text{thermal}}(T) = A \exp\left(aT - bT^2\right), \tag{1}$$

where the parameters classically proposed are the following: $b = 0$ and $a = 0.0423 \ {}^\circ\text{C}^{-1}$ based on North Atlantic seawater data (Takahashi et al., 1993); or $a = 0.0433 \ {}^\circ\text{C}^{-1}$ and $b = 8.7 \cdot 10^{-5} \ {}^\circ\text{C}^{-2}$ from a more recent work using a larger dataset (Takahashi et al., 2009). In the present work, we use the latter relation, but we note that the quadratic term is only providing slight corrections to the linear term. The coefficient $A$ is a multiplicative factor that has no influence on the dynamical or

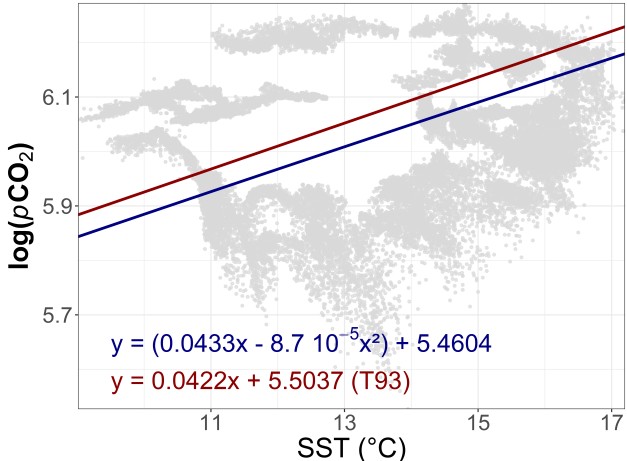

**Figure 2.** Distribution of the natural logarithm of the $p\mathrm{CO}_2$ data (in $\mu$atm) versus the SST (gray points). The empirical linear relation found by Takahashi et al. (1993, T93) is represented by the red line and the red equation. The equation used in this study is represented by the blue line and the blue equation, based on Takahashi et al. (2009).

statistical properties of $p\mathrm{CO}_2{}^{\text{thermal}}$. We estimate this value based on the data, as:

$$A = \left\langle \frac{p\mathrm{CO}_2(t)}{\exp\left(aT(t) - bT(t)^2\right)} \right\rangle, \tag{2}$$

where $\langle \cdot \rangle$ means statistical average, performed on the different values in the time series, where $t$ is the time. Figure 2 presents the value of $A$ derived in this study, along with the regression between SST and $\log(p\mathrm{CO}_2)$, compared with *in situ* measurements

105 and the regression reported by Takahashi et al. (1993). When considering the oxygen dynamics in marine waters, one classically divides the oxygen concentration value by the equilibrium oxygen which is a function of temperature and salinity: the ratio is a number without dimension, which is called the oxygen saturation percentage. Inspired by this approach, we consider here the non-thermal dynamics as the ratio $X_{\text{sat}}$ between the observed $p\mathrm{CO}_2$ and the thermal component $p\mathrm{CO}_2{}^{\text{thermal}}$:

$$X_{\text{sat}} = \frac{p\mathrm{CO}_2}{p\mathrm{CO}_2{}^{\text{thermal}}} \tag{3}$$

110 This provides a new time series, where $X_{\text{sat}}(t)$ is a non-dimensional quantity, indicating the saturation of $p\mathrm{CO}_2$; it can be seen as a percentage, and when it is larger than 1 (or 100 %) there is supersaturation, whereas when smaller than 1 there is undersaturation (with respect with an equilibrium value which depends on the temperature).

Let us note that a so-called "non-thermal" component of $p\mathrm{CO}_2$ is often introduced as (Takahashi et al., 1993, 2009; Wimart-Rousseau et al., 2020):

115 $$p\mathrm{CO}_2{}^{\text{non-thermal}} = p\mathrm{CO}_2 \times \frac{p\mathrm{CO}_2{}^{\text{thermal}}(T_{\text{mean}})}{p\mathrm{CO}_2{}^{\text{thermal}}(T)}, \tag{4}$$

where $T_{\text{mean}}$ is the average value of the temperature estimated for a given observation experiment. It can easily be shown using this definition and Eq. (3) that we have $p\mathrm{CO}_2{}^{\text{non-thermal}} = \lambda X_{\text{sat}}$ with a proportionality coefficient given by $\lambda = p\mathrm{CO}_2{}^{\text{thermal}}(T_{\text{mean}})$,

indicating that the classical expression of the non-thermal $p\text{CO}_2$ is proportional to the saturation value $X_{\text{sat}}$ which is considered in the present work. Their dynamics are thus the same, but we prefer here to consider the quantity $X_{\text{sat}}$ since it is a non-dimensional quantity and its comparison with the value of 100 % directly indicates supersaturation or undersaturation.

## 2.3 Fourier spectral analysis

Fourier spectral analysis is used to estimate the spectral dynamics of stationary time series. Due to missing data, the classical Fast Fourier Transform (FFT) algorithm, which needs regularly-space data, cannot be used. We therefore use here the Wiener-Khintchine theorem expressing that the spectral energy density is the Fourier transform of the autocorrelation function $C(\tau) = \langle X(t)X(t+\tau)\rangle$:

$$E(f) = \int_{-\infty}^{+\infty} C(\tau)\exp(-2i\pi\tau f)\mathrm{d}\tau, \tag{5}$$

where $f$ is the frequency, and $E(f)$ is the Fourier spectral energy density. The autocorrelation function can be directly estimated on time series with missing values, hence this algorithm is recommended for estimating the spectral dynamics of such time series. We consider here the code described in Gao et al. (2021). The power spectral density is represented in $\log$-$\log$ plot in order to emphasize possible scaling regimes of the form $E(f) \sim f^{-\beta}$, where $\beta > 0$ is the spectral exponent. For Brownian motion, $\beta = 2$, and for passive-scalar turbulence (i.e., a scalar that is advected by the flow but does not influence its dynamics) $\beta = 5/3$ (Kolmogorov, 1941; Obukhov, 1949; Corrsin, 1951).

## 2.4 Time-reversal symmetry

The statistical analysis of time-reversal symmetry allows to assess whether a time series exhibits the same statistical properties when considered in the forward direction of time or in reverse (i.e., by replacing $t$ with $-t$). A time series is considered reversible if its statistical properties remain unchanged regardless of the direction of time. This property was recently studied in the context of turbulence by Schmitt (2023), who observed an increase in time-reversal asymmetry within the inertial turbulent range. This is applied here to *in situ* ocean scalar data, and a time-reversal asymmetry indicator using triple correlations, proposed in the field of statistical physics by Pomeau (Pomeau, 1982, 2004), is used here. This indicator uses the following two triple correlations estimated for a stationary time series $u$:

$$G^+(\tau) = \frac{\langle u(t)u(t+2\tau)u(t+3\tau)\rangle}{g_3};$$
$$G^-(\tau) = \frac{\langle u(t+3\tau)u(t+\tau)u(t)\rangle}{g_3}, \tag{6}$$

where $u(t)$ is a zero-mean time series, and $g_3 = \langle u^3\rangle$ is a normalization, so that $G^+$ and $G^-$ are dimensionless quantities. The indicator $Po = |G^+ - G^-|$, denoted as the Pomeau index (Schmitt, 2023), quantifies the degree of time-reversal symmetry in a series. For a reversible series, where the direction of time does not affect the triple correlations, $Po = 0$. Conversely, the larger the value of $Po$, the more the series is asymmetric by time-reversal. Since $Po = 0$ for a reversible series, this indicator is also

interpreted here as an indicator of irreversibility in the time series. It can be evaluated for different values of the time increment $\tau$, allowing one to examine how irreversibility varies across scales.

## 2.5 Conditional mean

Conditional means are valuable exploratory tools. They provide a way to estimate the overall, often nonlinear, dynamics of a variable $u$ with respect to another variable $v$ (Wand and Jones, 1994):

$$m(x) = \langle v \,|\, u = x \rangle, \tag{7}$$

For empirical time series, an interval of values $(a, b)$ can be used instead of a fixed value $x$ in order to estimate the conditional mean:

$$m(x) = \langle v \,|\, u \in [a, b] \rangle, \text{ with } x = \tfrac{1}{2}(a + b) \tag{8}$$

This approach allows one to capture trends in the conditional dynamics between two variables which can be nonlinear, and cannot, by definition, be captured by a linear regression. The conditional average is usually computed at a given scale (here, over $[a, b]$), so that smaller-scale fluctuations are filtered.

## 2.6 PDF quotient $Q$

To analyze the relationship between two random variables, we complement the classical approach of conditional averaging with the Probability Density Function (PDF) quotient method. This method, introduced by Xu et al. (2007), involves estimating the following quotient:

$$Q(x, y) = \log_{10} \left( \frac{P_{u,v}(x, y)}{P_u(x) P_v(y)} \right), \tag{9}$$

where $P_u$ and $P_v$ are the respective PDF of the random variables $u$ and $v$, and $P_{u,v}(x, y)$ is their joint PDF. This quantity provides insights into the dependence between the two variables in the probability space. Indeed, if $u$ and $v$ are independent, $P_{u,v}(x, y) = P_u(x) P_v(y)$ and so $Q(x, y) = 0$. On the contrary, $Q$ is different from the zero value if a dependence exists. More precisely, $Q > 0$ indicates a positive relation, for which the joint probability density $P_{u,v}(x, y)$ is larger than the product $P_u(x) P_v(y)$ corresponding to the independence between the random variables. Conversely, the values for which $Q < 0$ indicate a "negative" relation, for which the joint probability density is smaller than what would be obtained if the variables were independent.

This provides information that complements what is obtained from the correlation. While the correlation is a single numerical value that quantifies the dependence between two variables for all their values (large or small), the PDF quotient reveals which specific values in the $(x, y)$ plane exhibit strong or weak statistical relationships between the variables.

# 3 Results

## 3.1 Statistical properties of $p\text{CO}_2^{\text{thermal}}$ and $X_{\text{sat}}$ in a turbulence framework

### 3.1.1 Means and coefficient of variation

The dataset analyzed in this study consists of the original variables given in Table 1 together with the computed series $p\text{CO}_2^{\text{thermal}}$ and $X_{\text{sat}}$. Hence we consider seven key oceanic scalars: the SST, SSS, chl $a$, $\text{O}_{\text{sat}}$, $p\text{CO}_2$ and associated thermal ($p\text{CO}_2^{\text{thermal}}$) and non-thermal ($X_{\text{sat}}$) components. First, basic statistical metrics were extracted from these time series, including the mean, standard deviation, and coefficient of variation ($CV$). These values are presented in Table 2. Most time series exhibit relatively high variability, ranging from 9 % to 17 %. However, three time series fall outside this range. Specifically, SSS and $\text{O}_{\text{sat}}$ show lower variability, with $CV$ values of 0.6 % and 4.7 %, respectively. In contrast, the chl $a$ time series possesses a much higher $CV$ of 55.6 %, largely due to its strong biological seasonality. Its mean concentration is relatively high, close to 1 $\mu\text{g L}^{-1}$, which is sometimes observed in highly mesotrophic waters (Simboura et al., 2005). The $CV$ of $p\text{CO}_2$ in this study is smaller than that observed for coastal ocean sites ($> 20$ %) in our previous work (Robache et al., 2025). Nevertheless, it is noteworthy that the $X_{\text{sat}}$ time series still exhibits noticeable fluctuations, even in this type of environment.

The database was introduced and analyzed in Gac et al. (2020), where various factors influencing the $p\text{CO}_2$ within the WEC were identified. These include tidal patterns, daily cycles, specific rainfall events, and biological processes such as respiration and photosynthesis. Expanding on this analysis, the study highlighted the multiscale variability of partial pressure of $\text{CO}_2$ in seawater, spanning timescales from six-hour fluctuations to seasonal trends. Based on these results, here we apply other metrics to characterize the dynamics and inter-relations of these time series.

**Table 2.** Means ($\mu$), standard deviations ($\sigma$) and coefficients of variation ($CV = \sigma/\mu$) for all the time series used in this study.

| Scalar (unit) | $\mu \pm \sigma$ | $CV$ (%) |
|---|---|---|
| $p\text{CO}_2$ ($\mu$atm) | $416.7 \pm 50.1$ | 12.0 |
| $X_{\text{sat}}$ (%) | $101.0 \pm 14.4$ | 14.3 |
| $p\text{CO}_2^{\text{thermal}}$ ($\mu$atm) | $409.3 \pm 37.1$ | 9.1 |
| Chl $a$ ($\mu\text{g L}^{-1}$) | $0.9 \pm 0.5$ | 55.6 |
| SST (°C) | $13.1 \pm 2.2$ | 16.8 |
| SSS (psu) | $35.2 \pm 0.2$ | 0.6 |
| $\text{O}_{\text{sat}}$ (%) | $102.2 \pm 4.8$ | 4.7 |

### 3.1.2 Presentation and distribution of the $X_{\text{sat}}$ time series

The time series of $X_{\text{sat}}$ is shown in Fig. 3. Notably, this variable does not remain consistently at 100 %, but exhibits fluctuations across various timescales, ranging from hours to years. Values vary from a minimum of 59.4 % in June 2016 to a maximum

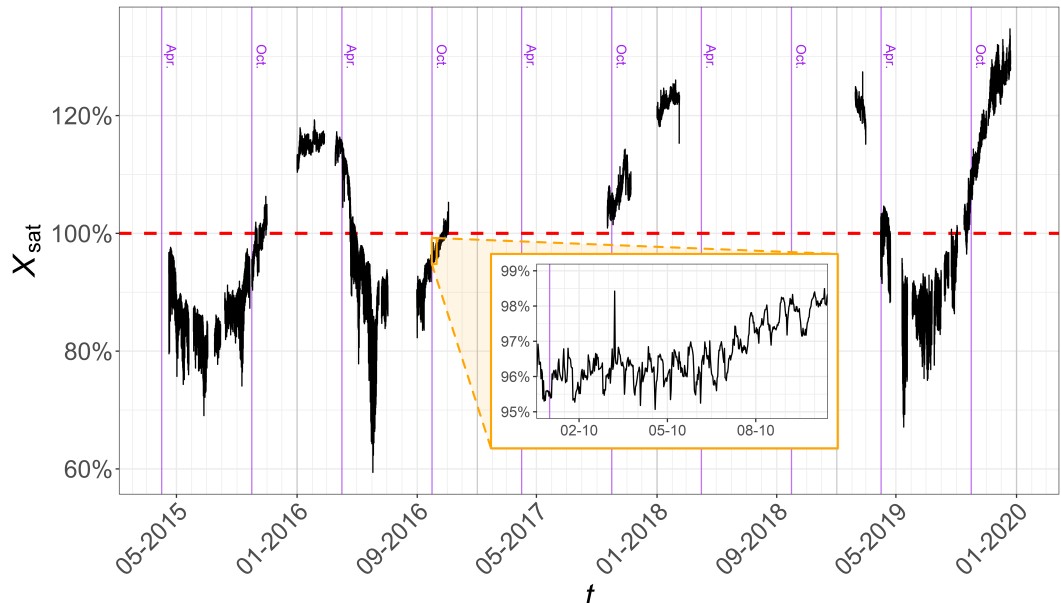

**Figure 3.** The complete $X_{sat}$ time series, as defined in Eq. (3), represented as a percentage. The red dashed line represents the 100 % value which is reached when $pCO_2 = pCO_2^{thermal}$. The inset is a zoom over a period of 10 days showing that the series presents fluctuations even at small scales. Purple horizontal lines indicate April and October of each year, providing quick visual reference points along the $x$-axis, which is spaced at monthly intervals.

of 134.7 % in December 2019. The mean value is 101.0±14.4 % with a variation coefficient of 14.3 %. Overall, this indicates
that the non-thermal component of $pCO_2$ exhibits non-negligible fluctuations, reflecting its own dynamics. At large scale, the
values seem to present a seasonal dynamics: $X_{sat}$ is larger than 100 % from October 2015 to April 2016, from September 2017
to February 2018, and from September to December 2019. As already mentioned in Gac et al. (2020), these super-saturated
values in winter could be linked with riverine inputs (e.g., from Morlaix and Penzé rivers) and to bacterial respiration. On the
contrary, undersaturated $X_{sat}$ values (smaller than 100 %) are found from March to October 2015, from April to October 2016,
and from April to September 2019: this corresponds to consumption of $CO_2$ by the primary production in spring and summer.
The PDF of $X_{sat}$ shown in Fig. 4a reflects this seasonal behavior, with peaks between 80 % and 90 % corresponding to primary
production, and above 110 % for winter values. Both Figs. 4a and 4b in linear and log-linear plots respectively show that the
PDF is non-Gaussian and also non-symmetric. More values and a longer tail were recorded below 100 %: approximately 55 %
of the values are below this threshold according to Fig. 4c. However, this result should be interpreted with caution due to the
presence of missing data in the series.

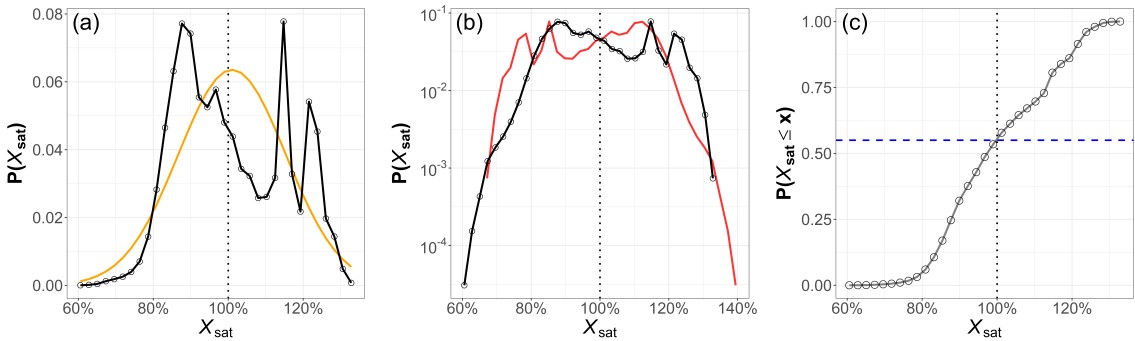

**Figure 4.** The PDF of $X_{\text{sat}}$ time series (%): (a) in linear plot, (b) in log-linear plot and (c) the distribution function. The orange line in (a) shows a Gaussian fit with the same mean and variance; the red line in (b) represents the symmetric PDF; the vertical dashed line in (c) denotes the 100 % value, reached when $p\text{CO}_2 = p\text{CO}_2^{\text{thermal}}$ and the horizontal blue line represents the probability $P(X_{\text{sat}} \leq 100) = 0.55$. The band width used in (a) and (b) corresponds to 3 % of the difference between the maximum and minimum values of $X_{\text{sat}}$.

### 3.1.3 Scaling properties

Fourier spectra have been estimated for each series and displayed in Fig. 5 in log-log plots, with the aim of characterizing the dynamics of the time series of the different scalars across temporal scales. Approximate scaling behavior are found visually; in order to estimate precisely the power-law slopes, power-law regressions (i.e., $y = x^{\beta}$) are estimated over the same range

of scales for each series, from 1.1 (to avoid the daily cycle influence) to 80 days. The highest frequencies (below the daily scale) are not considered to avoid the effects of missing data and the daily and tidal periodicities, which can disrupt the scaling properties (Schmitt and Huang, 2016). The values of $\beta$ and their regression standard deviation area provided in Table 3. In most cases the power-law fits estimated over this range of frequency scales extend approximately over low frequencies until the annual time scale. Additionally, distinct peaks were observed linked to tidal cycles ($\approx$ 12.4 hours in the WEC), and the

daily cycle (especially for chl *a*, Fig. 5d), and annual patterns.

All the slopes are close to the value of 5/3 corresponding to passive-scalar turbulence, with values ranging between 1.73 and 1.85. The shapes of the $p\text{CO}_2^{\text{thermal}}$ (Fig. 5c) and SST (Fig. 5e) spectra appeared very similar, with the same scaling exponents. The scalars which are usually defined as passive scalars (i.e. SST and SSS; Corrsin, 1951) possess here the smallest $\beta$ values: the spectral slopes for $X_{\text{sat}}$, $p\text{CO}_2$, $\text{O}_{\text{sat}}$ and chl *a* time series are closer to 1.80. This difference with the theoretical

5/3 value could be explained by a strong small-scale (e.g., from hours to months; Robache et al., 2025) intermittency for these scalars (Frisch, 1995; Seuront et al., 1996). In the turbulence framework, intermittency is an intrinsic property of turbulent time series (Frisch, 1995). Intermittency in turbulence was first revealed through early experimental measurements (Batchelor and Townsend, 1949): the term refers to strong, localized fluctuations in the velocity field, as well as in turbulent scalars. More broadly, this means that extreme fluctuations can be observed across different scales, occurring much more frequently

than would be expected for a Gaussian process. To assess this potential effect, we estimate the fraction of small-scale variance arising from fluctuations below the 12.4 hour tidal scale, using a 12.5-hour moving average. This smoothing procedure reduced

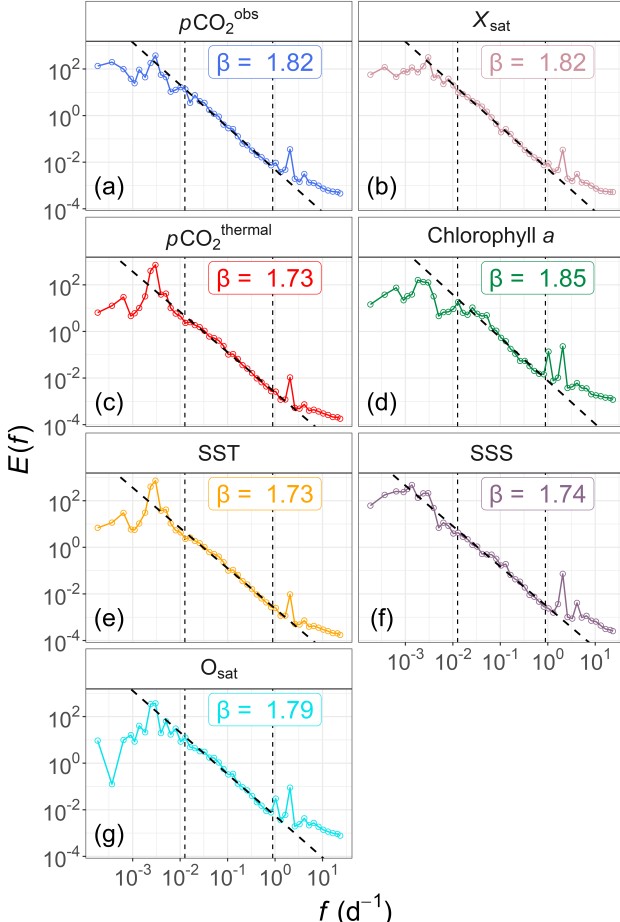

**Figure 5.** Fourier power spectral analysis, in log-log plots, of seven time series from the Astan buoy: (a) $pCO_2$, (b) $X_{sat}$, (c) $pCO_2^{thermal}$, (d) chlorophyll $a$, (e) SST, (f) SSS and (g) $O_{sat}$. The frequencies are expressed in day$^{-1}$. The black dashed lines represent the power-law regression for each time series, estimated for frequencies from 1.1 to 80 days. More information about the slopes $\beta$ associated to these spectra is given in Table 3.

high-frequency variability, effectively removing structures with periods shorter than 12.4 h. We then quantified the variance associated with these high-frequency fluctuations by computing the ratio between the variance of the smoothed ($\sigma^2_{smoothed}$) and original ($\sigma^2$) time series, expressed as a percentage:

$$\rho = 1 - \frac{\sigma^2_{smoothed}}{\sigma^2} \tag{10}$$


For small values of $\rho$, the contribution of small-scale variability is limited, whereas larger values of $\rho$ indicate a greater proportion of small-scale fluctuations. As shown in Fig. 6, the steepness of the slope $\beta$ may be related to the indicator $\rho$, showing that a relationship emerges between this index of small-scale variability and the value of $\beta$.

**Table 3.** Spectral slopes $\beta$ estimated for the seven considered parameters, for scales from 1 to 80 days. The associated spectra are displayed in Fig. 5. Difference between each slope values $\beta$ and $\beta_{\mathrm{SST}}$ are also given to quantify the difference with the passive scalar slope.

| Scalar | $\beta$ | $\beta - \beta_{\mathrm{SST}}$ |
|---|---|---|
| $p\mathrm{CO}_2$ | $1.82 \pm 0.05$ | 0.09 |
| $X_{\mathrm{sat}}$ | $1.82 \pm 0.04$ | 0.09 |
| $p\mathrm{CO}_2^{\,\mathrm{thermal}}$ | $1.73 \pm 0.05$ | 0 |
| Chl $a$ | $1.85 \pm 0.08$ | 0.12 |
| SST | $1.73 \pm 0.04$ | 0 |
| SSS | $1.74 \pm 0.05$ | 0.01 |
| $\mathrm{O}_{\mathrm{sat}}$ | $1.79 \pm 0.05$ | 0.06 |

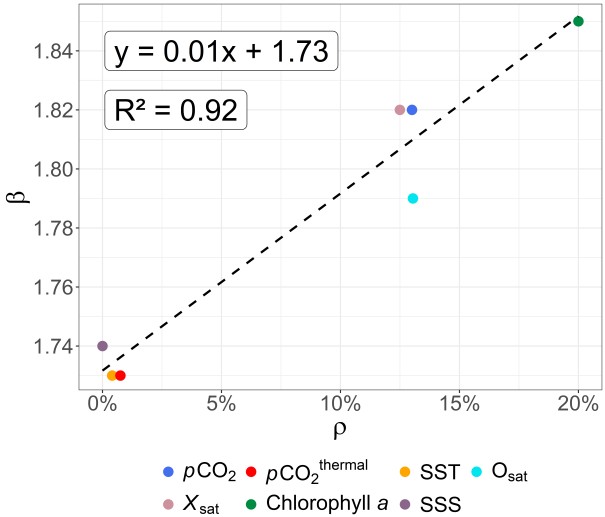

**Figure 6.** Fourier slope $\beta$ versus the indicator $\rho$, representing the influence of high-frequency fluctuations below tidal timescale for each time series. The black dashed line represents a linear regression.

### 3.1.4 Reversibility

Based on the results reported from the Fourier spectral analysis, the dynamics of each of the scalars considered in this study is considered using tools from the field of turbulence. Recently, the property of irreversibility—essential for characterizing the temporal symmetry of signal statistics—has been investigated in relation to turbulence (Schmitt, 2023). Indeed, in nature many phenomena can be modeled by a nonlinear relaxation dynamics, characterized by a rapid growth phase followed by a slower decay. Examples include epidemic outbreaks (Bestehorn et al., 2022), river discharge (Mathai and Mujumdar, 2022),
or El Niño–Southern Oscillation (ENSO; An and Jin, 2004) variability in climatology. The characterization of time-reversal

asymmetry in oceanography is thus of relevance in this context. To our knowledge, this type of analysis has never been applied to *in situ* environmental time series, and the turbulent context here offers a valuable opportunity to explore this property for the series under consideration. Therefore, the reversibility analysis has been applied to all seven time series. An example of triple correlation function $G$ is displayed in Fig. 7 for the SST series only, as an illustration: it is seen that the two functions $G^+(\tau)$

and $G^-(\tau)$ are close for scales $\tau$ below 25 days, and after this time scale, the two curves diverge quickly. The corresponding indicator *Po* has low values below this threshold, indicating a time reversibility for this temporal range, and irreversibility after 25 days, as shown in Fig. 8. In this figure all values of $Po(\tau)$ for scales from 0 to 70 days are superposed, since this indicator is dimensionless: it corresponds to a quantification of the departure from 0 of the time-reversal symmetry of each series. We can see that for three series, the curves stays very close to zero at all scales: SSS, chl *a* and $O_{sat}$. These series may be considered as

time-reversal symmetric. For the other series, for scales below 30 days, the indicator remains very small, with values smaller than 2, an arbitrary threshold we have plotted in the figure. It is then increasing and reaching an approximate plateau (with fluctuations) for scales from 50 to 70 days.

We then consider in Table 4 several values derived from the $Po(\tau)$ curves: the time scale $\tau_{min}$ corresponding to the first time for which $Po(\tau) > 2$ (for three series this value cannot be extracted since the curve always remains close to zero); the

maximal value $Po_{max}$ reached by *Po*; and $\overline{Po}$, the average value of *Po* estimated over scales $\tau \in [50; 70]$. This shows that the four series $pCO_2$, $X_{sat}$, $pCO_2^{thermal}$ and SST become clearly irreversible for timescales larger than 30 to 50 days, the larger value $\tau_{min} = 50.7$ is for the $X_{sat}$ time series. Further, $Po_{max}$ as well as $\overline{Po}$ are the larger for the $pCO_2$ time series, followed by $pCO_2^{thermal}$, SST and $X_{sat}$ in decreasing order. As for the Fourier spectra, it can be seen that SST and $pCO_2^{thermal}$ *Po* time series dynamics are very similar.

In summary, there are two types of dynamics: chl *a*, SSS and $O_{sat}$ are found to be time-reversal symmetric, whereas the other series are time-reversal symmetric for scales below one month, and become irreversible for larger scales, the stronger irreversibility being found for $pCO_2$ and $pCO_2^{thermal}$.

**Table 4.** The minimum value $\tau_{min}$ for which $Po > 2$; the maximum value of *Po*, and the mean $\overline{Po}$ and standard deviation of *Po* estimated for $\tau \in [50; 70]$ for each scalar.

| Scalar | $\tau_{min}(Po \geq 2)$ (days) | $Po_{max}$ | $\overline{Po} \pm \sigma$ |
|---|---|---|---|
| $pCO_2$ | 34.2 | 28.0 | $20.9 \pm 4.9$ |
| $pCO_2^{thermal}$ | 31.6 | 9.9 | $8.8 \pm 0.5$ |
| SST | 36.1 | 5.3 | $4.7 \pm 0.3$ |
| $X_{sat}$ | 50.7 | 3.0 | $2.5 \pm 0.5$ |
| Chl *a* | - | 0.2 | - |
| SSS | - | 0.3 | - |
| $O_{sat}$ | - | 0.4 | - |

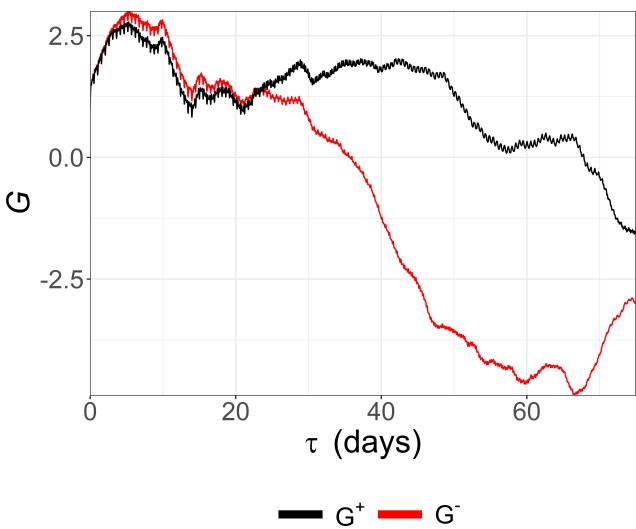

**Figure 7.** Triple correlation functions $G^+(\tau)$ (black line) and $G^-(\tau)$ (red line) for the SST time series.

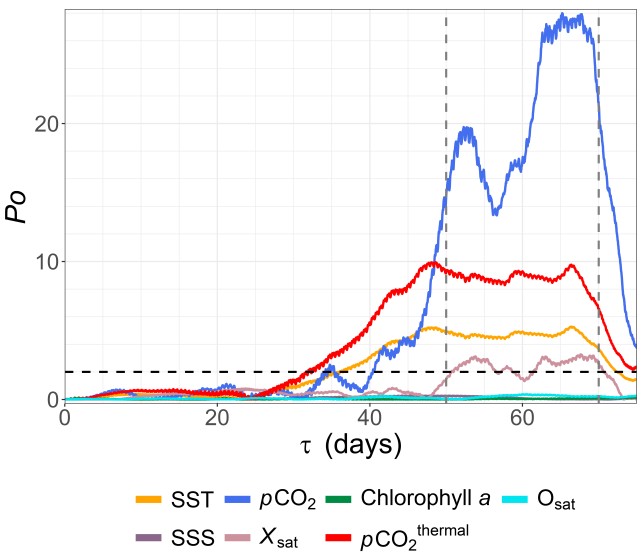

**Figure 8.** The indicator $Po$ versus $\tau$ for all seven time series. The horizontal dashed line represents the value $Po = 2$. The vertical dashed lines represent the values $\tau = 50$ and $\tau = 70$ days.

### 3.2 Multivariate statistics

In the previous sections, we focused on the univariate statistical properties of the scalars. In this section, we turn to the statistical relationships that may exist between different scalars, using a range of methods, some more common than others, with the aim of characterizing them.

#### 3.2.1 Conditional means

In this subsection and the following the focus is directly on the behavior of $X_{\text{sat}}$. We consider here first the conditional means of $X_{\text{sat}}$ with respect to the values of other variables, to visualize the interactions between variables. They are displayed in Fig. 9. First, three plots show a direct and clear relation. There is as expected a direct relation between $p\text{CO}_2$ and $X_{\text{sat}}$ visible in Fig. 9d. Further, a clear relation is also observed between the chl $a$ concentration and $X_{\text{sat}}$ in Fig. 9e: an increase in the former is linked with a decrease in the latter. The same relation is observed in Fig. 9f, suggesting that it may be the effect of primary production.

Furthermore, a link between $X_{\text{sat}}$ and SSS is also noted, however less clear for large values of SSS, in Fig. 9b: low salinity values are linked to high values of $X_{\text{sat}}$. A similar relationship is found with atmospheric pressure $P_{\text{atm}}$ in Fig. 9c: large values of $X_{\text{sat}}$ are found in case of atmospheric depressions. These two observations suggest an effect of rainfall events and/or river discharges, which impact $p\text{CO}_2$. This quantitatively confirms what was visually mentioned in Gac et al. (2020).

Finally, a link is also found between SST and $X_{\text{sat}}$ in Fig. 9a. This observation may seem counterintuitive due to the definition of $X_{\text{sat}}$, in which $p\text{CO}_2$ is corrected from direct temperature influences. However, the pattern in Fig. 9a can be explained by indirect influences: on the one hand, there is the annual seasonality of SST and on the other hand, there are different seasonal processes impacting $p\text{CO}_2$ without having a direct link with temperature dynamics, such as phytoplankton blooms during spring or notable rainfall events and storms during autumn and winter. Moreover, this association is primarily linked to the seasonality of biological activity, as demonstrated in Fig. 10. Indeed, the two time series presented in Fig. 10a appear to have an opposite dynamics: their Pearson correlation yields a value of -0.84 (with a p value $\approx 1\ 10^{-12}$). The monthly seasonality is also depicted in Fig. 10b, where the two time series appear to have seasonal averages that are clearly related through a seasonal elongated lemniscate (i.e., a curve shaped like a figure eight or an infinity symbol).

#### 3.2.2 PDF quotient $Q(x, y)$

In this subsection, the statistical relations of $X_{\text{sat}}$ with other parameters is considered using the PDF quotient method. As explained in Sect. 2.6, the correlation (or covariance) between two series provides only a single number, without accounting for how both variables may be related across their range of variation. Here, the PDF quotient makes it possible to identify the ranges of values over which the scalars are more or less dependent on each other, and the strength of this dependence. The results are presented in Fig. 11, with blue and red colors for negative and positive values of $Q(x, y)$, respectively. Let us recall here that regions for which $Q > 0$ indicate values for which there are strong statistical relationships between the variables and when $Q = 0$ there are no statistical relations between the variables. Low values of Q are emphasized by gray

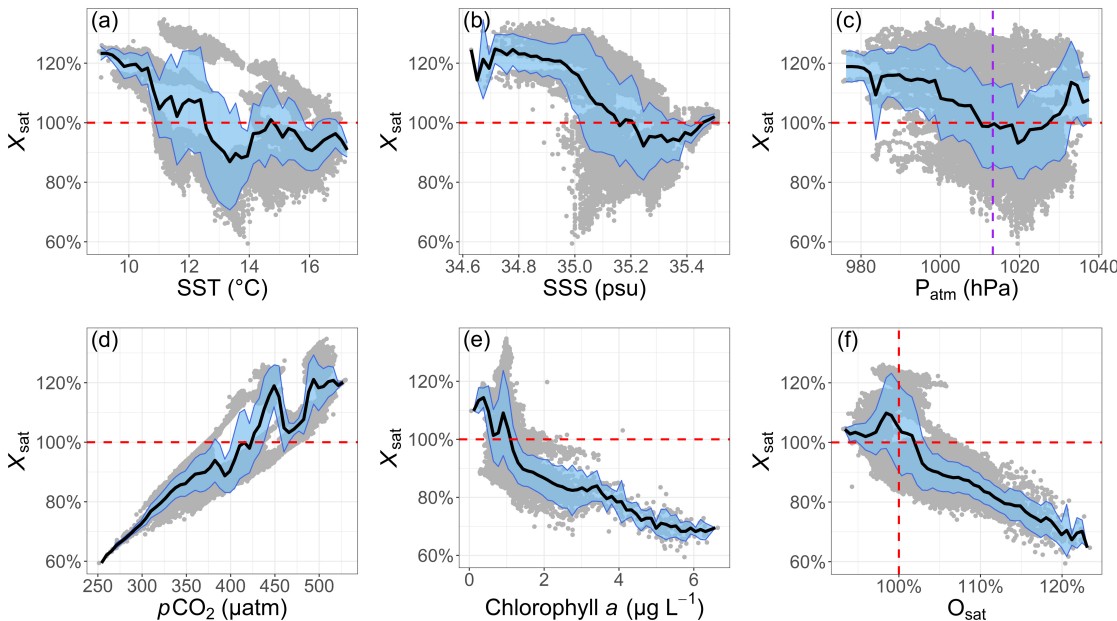

**Figure 9.** Conditional means of $X_{sat}(t)$ with respect to several other variables: (a) SST, (b) SSS, (c) atmospheric pressure $P_{atm}$, (d) $pCO_2$, (e) chlorophyll $a$ and (f) oxygen saturation $O_{sat}$. The blue area represents the standard deviation calculated for each range. The red vertical dashed lines—and the horizontal one in panel (f)—represent the 100 % value. The purple vertical dashed line in panel (c) represent the mean atmospheric pressure at mean sea level (1013.25 hPa). For each variable, the chosen bandwidth $b_w$ used for the calculation of the conditional mean is defined as 2 % of the difference between its maximum and minimum values. The $x$-value is the mean of each bounds of each range.

zones (for which $|Q| < 0.3$). They correspond to an lack of statistical relationship as found for a large part of the panel (c) for atmospheric pressure, or of panel (b) for SSS. On the contrary red values show large relationships. Table 5 shows the minimum and maximum values of the PDF quotient for each parameter. This range shows that the minimum values are close for each parameter, and that the maximum values is much larger for $pCO_2$, chl $a$ and $O_{sat}$ (with values from $10^{4.1} \simeq 12\,600$ to $10^{4.5} \simeq 31\,600$) than for SST, SSS or $P_{atm}$ (with values from $10^{2.1} \simeq 125$ to $10^{2.3} \simeq 200$). Such large values indicate a strong statistical relationship. More comments are provided below as related with the values of $X_{sat}$.

We first consider the half-plane for which $X_{sat} < 100$ %: panel (e) shows that large $Q$ values are observed for large chl $a$ ($> 2\ \mu$g L$^{-1}$) and low $X_{sat}$ values ($< 85$ %), and panel (f) similarly that large $Q$ values are found for large $O_{sat}$ ($> 110$ %) and low $pCO_2$ ($< 350\ \mu$atm) values. This corresponded to phytoplankton bloom periods, during which low $X_{sat}$ values are reached due to $CO_2$ consumption associated with photosynthesis: the PDF quotient method is a way to provide a quantification that confirms visual observations. The links between chl $a$ and $X_{sat}$ are more complex for values below $2\ \mu$g L$^{-1}$, with $Q$ values closer to 0. In some cases, a "negative" relation ($Q < 0$) can even be found. More globally, an independency ($Q = 0$) or a lower relation ($|Q| < 1$) was more often observed for this range of chl $a$ values, which are more often recorded referring to the associated joint probabilities (97 % of the chl $a$ recorded values).

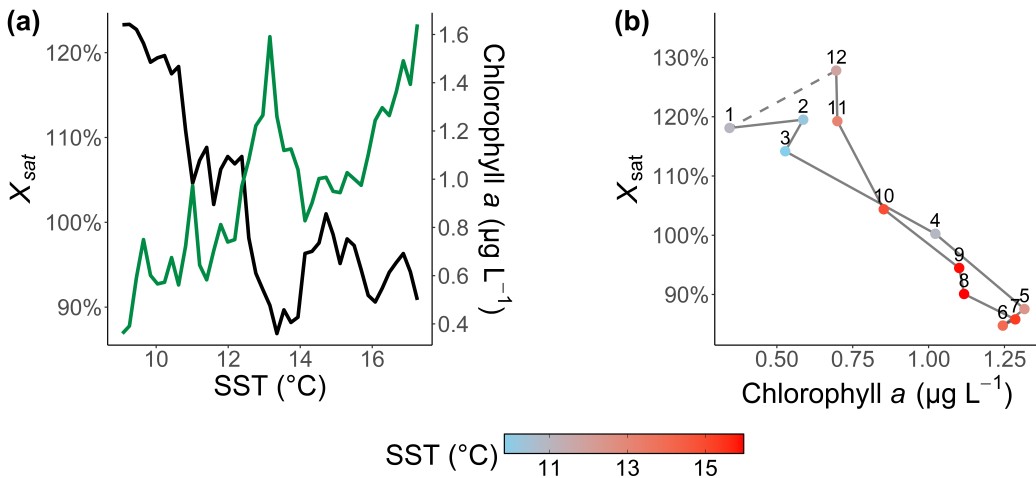

**Figure 10.** Illustration of the link between $X_{\text{sat}}$ and the seasonality of the chlorophyll $a$: (a) conditional means of chlorophyll $a$ (green line) and $X_{\text{sat}}$ (black line) versus the SST (as in Fig. 9) and (b) mean $X_{\text{sat}}$ versus the mean chlorophyll $a$ (black line). The color of the dots indicates the mean SST for each month of the year themselves indicated by numbers (1 for January, 12 for December).

For intermediate values of $X_{\text{sat}}$ (between 85 % and 115 %, representing 64 % of recorded data), the patterns are often mixed. The mean values of $Q$ for this $X_{\text{sat}}$ range of values were globally all smaller than 1 and even close to zero in some cases, e.g. with chl $a$ for $X_{\text{sat}}$ values ranging between 100 and 105 % (Fig. 11e). This is especially true between 85 % and 95 %, where no clear link is observed with any other scalar, visible by the gray zones in the figures. For $X_{\text{sat}}$ values between 95 % and 105 %, larger values of $Q$ are found for $O_{\text{sat}} < 95$ %.

Finally, let us consider large values of $X_{\text{sat}}$ ($> 115$ %), for which large $Q$ values are observed for the lower part of the SSS values (below 35 psu) in panel (b), and below 1000 hPa for the pressure in panel (c). The two observations are consistent and indicate that supersaturation of $X_{\text{sat}}$ corresponds to a depression with lower values of salinity due to rainfall or riverine inputs. In panel (a) the lower temperatures associated with large values of $Q$ indicate that such events happen in winter. Here also the PDF quotient approach helps to quantify relationships and to provide a quantitative justification of proposed mechanisms.

**Table 5.** The minimum and maximum values of the PDF quotient $Q$ for each scalar in comparison to $X_{\text{sat}}$.

| Scalar | $Q_{\min}$ | $Q_{\max}$ |
|---|---|---|
| SST | -1.3 | 2.1 |
| SSS | -1.4 | 2.2 |
| $P_{\text{atm}}$ | -1.2 | 2.3 |
| $p\text{CO}_2$ | -1.6 | 4.5 |
| Chl $a$ | -1.6 | 4.1 |
| $O_{\text{sat}}$ | -1.5 | 4.2 |

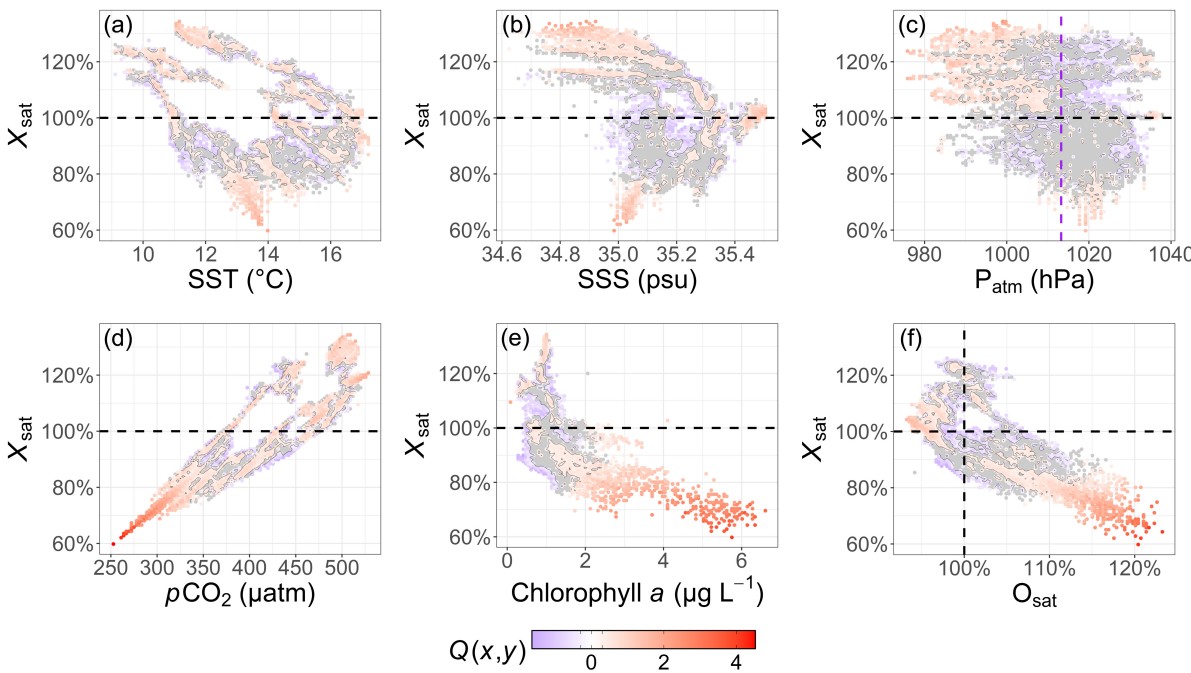

**Figure 11.** Representation of the PDF quotient $Q(x, y)$ estimated for $X_{sat}$ and (a) SST, (b) SSS, (c) atmospheric pressure $P_{atm}$, (d) $pCO_2$, (e) chlorophyll $a$ and (f) oxygen saturation $O_{sat}$. The horizontal black dashed lines—and the horizontal one in panel (f)—represent the 100 % value. The vertical purple dashed line in panel (c) represents the mean atmospheric pressure at mean sea level (1013.25 hPa). For each variable, the chosen bandwidth used for the calculation of the 2D-PDF is defined as 1 % of the difference between its maximum and minimum values. The x-value and y-value are the mean of each bounds of each range. A contour plot of the value $|Q| = 0.3$ is represented in each panel and the value below this threshold are represented in gray, to separate visually the area where $10^{|Q|} < 2$.

## 4 Discussion

The time series recorded on the Astan cardinal buoy have been used to study the statistical properties of several marine scalars, of which the thermal and non-thermal components of the seawater partial pressure of $CO_2$. First, a quantity, $X_{sat}$, was proposed in Eq. (3), similar to $pCO_2^{\text{non-thermal}}$ found in the literature (Takahashi et al., 1993, 2009; Wimart-Rousseau et al., 2020) but proportional to it. The advantage of using $X_{sat}$ is that it provides a $pCO_2$ saturation percentage, allowing for the direct identification of sources (supersaturation) or sinks (undersaturation). For the database studied here, $X_{sat}$ primarily ranges between 80 ($5^{th}$ percentile = 81 %) and 120 % ($95^{th}$ percentile = 124 %).

In this work, we focused on $X_{sat}$ and its relationship with other measured quantities. All marine quantities studied are influenced by turbulence, exhibiting scale-invariant statistics with a spectral exponent $\beta$ close to 5/3 (Fig. 5) over timescales ranging from 12 hours to at least 80-100 days. In certain time series, annual, daily, and especially tidal forcing were evident. For some series, the $\beta$ values were larger than the 5/3 Kolmogorov-Obukhov-Corrsin passive scalar value corresponding to

non-intermittent turbulence. This has already been found for SST and chl $a$ in a previous study in the eastern English Channel (Seuront et al., 1996). The slightly steeper slopes found for some series could be related with the intermittency, as supported by the link observed between the importance of the high-frequency fluctuations, below the tidal forcing timescale ($\simeq 12.4$ h) on the scalar dynamics and the associated $\beta$ values found (Fig. 6). Remarkably, the 5/3 slope was also observed for $X_{\text{sat}}$. This indicates that even without thermodynamic effects, oceanic $p\text{CO}_2$ dynamics in the western English Channel (WEC) remains influenced by turbulence. This result differs from our previous study, where we found a mean power-law slope of 1.47 for coastal seawater time series (Robache et al., 2025). This discrepancy could be attributed to the strong hydrodynamic forcing in the present dataset, primarily driven by the macrotidal dynamics of the WEC (tidal range $> 4$ m; Levoy et al., 2000; Idier et al., 2012).

Despite these similarities in spectral slopes, the time series exhibit different dynamics, as revealed by the reversibility analysis through the time reversal symmetry analysis. Interestingly, quantities more closely linked to biology, such as $\text{O}_{\text{sat}}$ and chl $a$, and even SSS, are reversible, despite their otherwise turbulent dynamics. Other quantities are reversible at small scales and become irreversible at timescales greater than 30 days. At these larger timescales, we found that the most irreversible signals are those of $p\text{CO}_2$ and $p\text{CO}_2^{\text{thermal}}$. The quantity $X_{\text{sat}}$ exhibits an intermediate behavior between SST, $p\text{CO}_2$, and the reversible variables, as if accounting for the direct effects of temperature had brought it closer to biological dynamic. The fact that some series become irreversible at scales larger than 30 to 50 days could be related to climate forcing. Indeed, irreversibility in climate time series has already been observed by Giancaterini et al. (2022). Moreover, we applied Empirical Mode Decomposition, an algorithm that extracts modal fluctuations from a time series (Huang et al., 1998), to SST (not shown here). This analysis revealed that irreversibility may be associated with large-scale modal fluctuations, including those with periods of one year or longer.

The relation between variables have been investigated for $X_{\text{sat}}$, with the conditional mean and Probability Density Function (PDF) quotient methods. The conditional means, and especially the PDF quotients, have made it possible to quantify the observation first made visually on the $X_{\text{sat}}$ series. For low values of $X_{\text{sat}}$, the conditional mean, as well as the PDF quotient, showed that these values are associated with high values of chl $a$ and $\text{O}_{\text{sat}}$, showing that they are associated with photosynthesis and primary production at temperatures between 12 and 14 °C. High $X_{\text{sat}}$ values are often associated with low atmospheric pressures, lower SSS values (visible through the PDF quotient) and, for some of them, lower temperature values ($< 12$ °C). This indicates that these supersaturation episodes are found during depressions associated with rainfall (atmospheric inputs) and rivers. Another interesting result was the finding of a indirect effect of SST on $X_{\text{sat}}$, due to the links between SST and other parameters influencing the non-thermal component of $\text{CO}_2$ such as chl $a$. The PDF quotient is a novel methodology that was proposed in the past (Xu et al., 2007), yet it has received limited utilization since its inception. In this study, we have employed the PDF quotient method in a systematic manner to augment the conventional bivariate conditional mean analysis. The PDF quotient approach has enabled us to identify the values in the $(x, y)$ space of the variables under investigation where the statistical relationship is most robust. This approach is also noteworthy for its ability to highlight areas in the $(x, y)$ space where statistical independence exists, i.e. values of $Q$ close to zero, as illustrated in gray in Fig. 11. For example, it is evident from Fig. 11 that atmospheric pressure shows no correlation with $X_{\text{sat}}$, except in one quadrant where $X_{\text{sat}}$ has high

values and simultaneously the pressure is low, as illustrated in panel (c). This also shows that salinity exhibits no relationship with $X_{\text{sat}}$, except in areas of low salinity and high $X_{\text{sat}}$ values, as shown in panel (b). Furthermore, it can also be seen that when $X_{\text{sat}}$ assumes slightly undersaturated values (between 90 and 100 %), all the figures corresponding to the PDF quotient associated with different quantities are grayed out, indicating that the PDF quotient approaches zero for these values of $X_{\text{sat}}$. This observation demonstrates that these slightly undersaturated non-thermal $p\text{CO}_2$ values exhibit no discernible relationship with any variable; the underlying causes are multifaceted, and no single parameter appears to be determinative. This is in contrast to the behavior observed for low or very high trans values of $X_{\text{sat}}$, as previously discussed.

The methodological framework used in this study highlights the importance of advanced statistical tools for analyzing fixed-point time series, particularly in highly non-linear and turbulent environments such as WEC. The combination of spectral, time-reversal symmetry, and conditional dependence (conditional means and PDF quotient) analysis has allowed us to study and better understand the complex and scale-dependent dynamics of several scalars and of their relation. The non-linear nature of oceanic $p\text{CO}_2$ dynamics is evident in the interplay between physical, chemical and biological processes, which influence the variability of $X_{\text{sat}}$ and $p\text{CO}_2^{\text{thermal}}$ in a non-linear way across different timescales. Nevertheless, given the 5-year length of the dataset and the presence of gaps, our analysis is best suited to characterizing sub-annual variability and its drivers. The statistics reported here may depend on the forcing associated with interannual variability and its potential anthropogenic forcing. Further analyses, performed over different periods of time, will be needed in future works, in order to compare with the results presented here. Also, longer continuous time series would be required to extend these analyses to larger temporal scales for which other dynamics could be expected (e.g., long-term memory at decadal timescale; Séférian et al., 2013). It is also necessary for these time series to be as temporally homogeneous as possible (i.e., with no or few interruptions) in order to enhance the robustness of the applied statistical analyses. Indeed, in the present case, although each time scale is represented by sufficient statistics, the data are not fully homogeneous on a monthly basis. Appendix A shows that December and March were undersampled compared with other months. At the seasonal scale, however, the distributions are more balanced. This temporal heterogeneity represents one limitation of our analysis and highlights the need for future comparisons with datasets containing fewer missing values. Appendix B examines the effects of missing data on the analysis of PDF quotients, restricting the computation to observations for which all parameters were simultaneously available. The influence of missing values is limited in this case, and the results are broadly consistent with those presented in the main analysis.

This study also underscores the necessity of considering non-linear dynamics and interactions when analyzing air-sea $\text{CO}_2$ fluxes and identifying source-sink dynamics, as the dynamics of $\Delta p\text{CO}_2$ is highly linked to oceanic $p\text{CO}_2$ (Robache et al., 2025). The emergence of different behaviors at various temporal scales—such as reversibility at small scales and increasing irreversibility beyond 30–50 days—suggests that deterministic and stochastic influences act differently across timescales in the WEC carbon cycle. This confirms the complex nature of oceanic carbon dynamics and fluxes and underscores the necessity of high-frequency, robust statistical analyses to fully capture and understand them.

# 5 Conclusions

Fixed-point measurements, collected via high-frequency buoys, are of critical importance for the study of coastal marine dynamics. They also provide a theoretical framework for interpreting fluctuations, as demonstrated here through the application of Eulerian turbulence. For this, high frequency is necessary in order to access the dynamics at small scales, a necessity due to the presence of strong multiscale fluctuations in the systems under study. In this context, a novel quantity, designated $X_{\text{sat}}$, was considered, with its properties evaluated in relation to the other variables. This quantity directly allows $pCO_2$ of the direct thermal part to be corrected, and highlights periods of supersaturation or undersaturation linked to non-thermal events. We were thus able to highlight here the influence of turbulence on the different parameters, including the $X_{\text{sat}}$ quantity, for scales between 1 day and 80-100 days. Our results also highlight a marked irreversibility in $pCO_2$ time series above 30 days, suggesting the influence of non-stationary and dissipative processes, likely linked to underlying oceanic and atmospheric dynamics. This temporal turbulent and asymmetric dynamics has implications for $pCO_2$ modeling and forecasting, emphasizing the need for approaches that account for irreversible fluctuations and turbulent transport mechanisms in the ocean and atmosphere.

In addition, we explored the potential of a little-known method, the PDF quotient, which makes it possible to identify values for which variables have statistical relationships between them. This enabled us to quantitatively analyze visual observations, such as the influence of primary production on the value of $X_{\text{sat}}$ (for $X_{\text{sat}}$ below 85 %), or the influence of periods of depression on supersaturation (for $X_{\text{sat}}$ above 115 %) due to atmospheric or terrigenous inputs. This method provided new insights into the stochastic coupling between biological and physical processes, reinforcing the importance of considering both drivers when interpreting high-frequency $pCO_2$ variability. Furthermore, this analysis confirms the complex relationship between $pCO_2$ and other variables considered as its drivers, such as chlorophyll $a$ which can serve as a proxy for the biological carbon pump, emphasizing the need for deeper investigations into their statistical causal relations.

Finally, the findings of this study have direct potential applications in oceanographic monitoring and carbon cycle research. The methodology developed here enhances the accuracy of oceanic $pCO_2$ dynamics analysis, providing a more precise characterization of high-frequency variability. This improved characterization contributes to a better understanding of coastal biogeochemical processes across multiple temporal scales—an essential step, for example, toward verifying and improving model-based forecasts.

## Appendix A: About the missing data problem

Most of the analysis methods employed in this study account for the issue of missing values, a common problem when working with high-frequency in situ observations. For instance, spectral analysis, time-reversal symmetry analysis, and the PDF quotient of increments all incorporate the temporal variability of $pCO_2$ by considering differences between observations separated by a given time scale $\tau$. This approach ensures that, despite occasional data gaps—sometimes extending over relatively long periods—a sufficient number of samples remains available at each time scale. Figure A1 illustrates the number of $pCO_2$ increments available for all considered scales between 30 minutes and 365 days. Despite approximately 60 % missing data in the time series (the highest rate among our variables; Table 1), more than 3900 temporal increments are available for timescales shorter than one year. This provides sufficient statistical support for our analyses, which rely predominantly on mean values and are restricted to these timescales.

On the other hand, the distribution of the available $pCO_2$ data over the five-year period, shown as a function of month and season in Fig. A2, is not perfectly homogeneous. Some months, such as December and March, are under-sampled relative to others, which likely results in an under-representation of their dynamics in the aggregated statistics. At the seasonal scale, the distribution is more balanced, although autumn contributes proportionally more data than the other seasons. This sampling structure should be considered when interpreting the results. A full assessment of its influence would require a continuous time series covering the entire period.

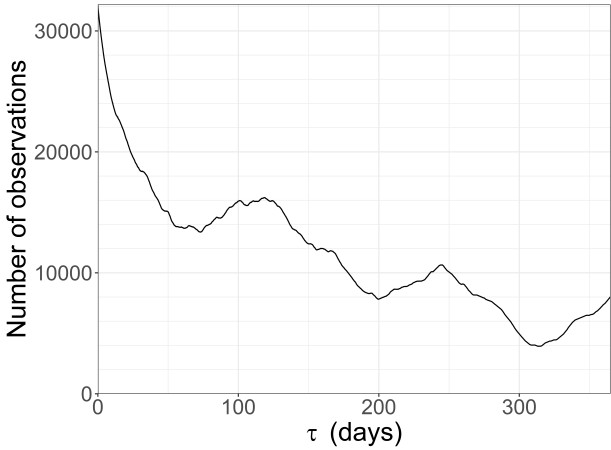

**Figure A1.** Number of observed temporal increments of $pCO_2$ as a function of the considered timescale $\tau$.

## Appendix B: Testing the effect of missing data on the PDF quotient analysis

In the analysis presented in Sect. 3.2.2, the numbers of value pairs used to estimate the PDF quotients are not all equal, due to the fact that the missing data are not the same for all series. To verify the robustness of our results, we therefore compared

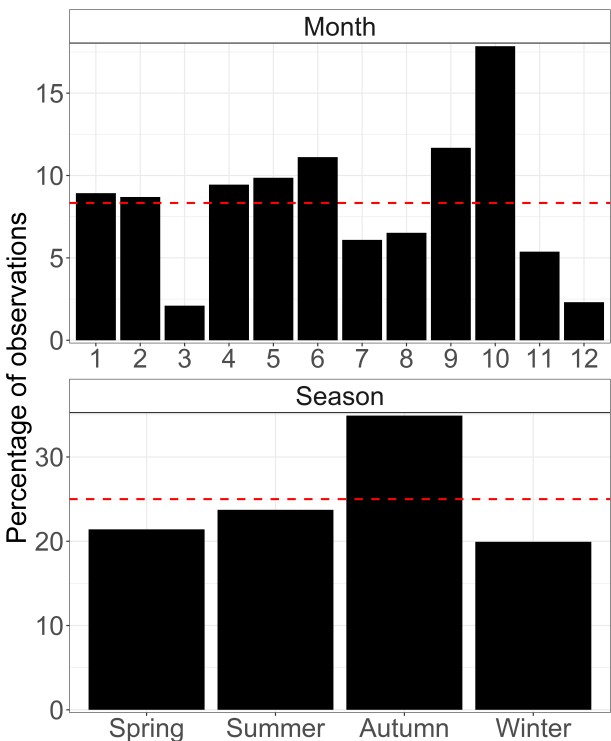

**Figure A2.** Distribution of the available $p\mathrm{CO_2}$ data by month and season. The red dashed line represents the expected values for a uniform sampling.

these results with those obtained by considering only the observations for which all parameters were simultaneously present (no missing data in any of the parameters). First, a comparison of the number of pairs used for each parameter in both cases is presented in Table B1. In the case where only simultaneous observations of all scalars are retained (case B in the table), the number of available observations decreases due to missing data for chl $a$ and $\mathrm{O_{sat}}$. These missing values do not occur at the same periods, which explains this reduction. We examined the monthly distribution of the available data pairs in each case. The results are shown in Fig. B1. Panel (a) corresponds to case A. For this case, the distribution of data pairs of SST, chl $a$, and $\mathrm{O_{sat}}$ with $X_{\mathrm{sat}}$ is provided. It is not shown for the other scalar variables since their distributions are similar to that of SST. Notably, in case A, most months are covered for all scalars except for $\mathrm{O_{sat}}$, for which December is missing. However, the temporal coverage is not completely uniform at the monthly scale. It is somewhat more consistent at the seasonal scale. For instance, during winter, the values from January and February allow winter conditions to be represented in the results. For case B, shown in panel (b), the distributions are overall similar to those obtained previously for chl $a$ and $\mathrm{O_{sat}}$, except for November and December, when very few values are retained due to the homogenization of the time series across the different scalars (as explained earlier).

**Table B1.** Number of value pairs used to estimate the PDF quotient (compared with $X_{sat}$ values) for each scalar, either by retaining all available pairs (case A) or by considering only the values for which simultaneous measurements of all scalars were available (case B).

| Scalar | Case A | Case B |
|---|---|---|
| SST | 32 582 | 18 292 |
| SSS | 32 582 | 18 292 |
| $P_{atm}$ | 32 582 | 18 292 |
| $pCO_2$ | 32 582 | 18 292 |
| Chl $a$ | 23 238 | 18 292 |
| $O_{sat}$ | 26 480 | 18 292 |

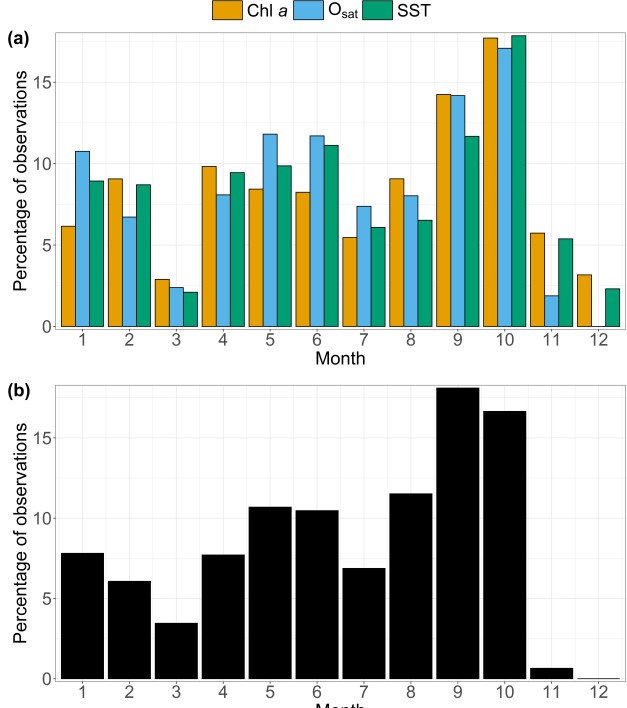

**Figure B1.** Percentage of observations (%) per month of the year, considering (a) all available $X_{sat}$ data (case A) and (b) only the subset of data with complete observations for all scalars (case B).

Following this, we repeated the PDF quotient analysis, this time considering only the values recorded simultaneously. The
results obtained for SSS and chl $a$ are presented in Fig. B2. Overall, the results from this analysis remain consistent with those obtained in Sect. 3.2.2. A relationship is still observed between high $X_{sat}$ values and SSS, as well as between high chl $a$ values

and low $X_{\text{sat}}$ values. The results for the other scalars are not shown here for readability, but the conclusions remain the same for each of them.

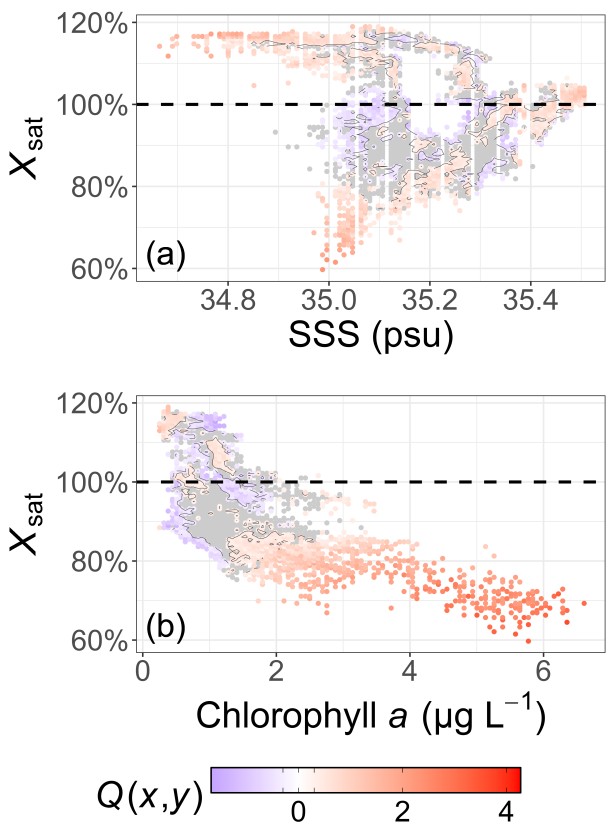

**Figure B2.** Representation of the PDF quotient $Q(x, y)$ estimated for $X_{\text{sat}}$ and (a) SSS and (b) chlorophyll $a$, using only the data for which all scalars were measured simultaneously (case B). The horizontal black dashed lines represent the 100 % value. For each variable, the chosen bandwidth used for the calculation of the 2D-PDF is defined as 1 % of the difference between its maximum and minimum values. The x-value and y-value are the mean of each bounds of each range. A contour plot of the value $|Q| = 0.3$ is represented in each panel and the value below this threshold are represented in gray, to separate visually the area where $10^{|Q|} < 2$.

Finally, by relying on a smaller number of observations, some important information is actually lost regarding the interaction
between SSS and $X_{\text{sat}}$ (and similarly for atmospheric pressure and SST), for which more values are available (i.e., all $X_{\text{sat}}$ values). For this reason, in this study we retained in the main text the analyses that included the largest number of values.

*Code and data availability.* The used database was provided by Gac et al. (2020), and is published in Bozec et al. (2025) and Robache and Schmitt (2025). The Python code for the Fourier spectral analysis is based on Gao et al. (2021): https://github.com/lanlankai/Scaling-Analysis-of-the-CFOSAT-Along-Track-Wind-and-Wave-Data (last access: 11 September 2025). The R code for the PDF quotient analysis has been published here: https://github.com/KevinRobache/PDF_Quotient_Code (last access: 11 September 2025). All the plots have been made using the 'ggplot2' (https://doi.org/10.32614/CRAN.package.ggplot2; Wickham et al., 2025), other 'tidyverse' (https://doi.org/10.32614/CRAN.package.tidyverse; Wickham et al., 2019; Wickham and RStudio, 2023), and 'ggmagnify' (https://github.com/hughjonesd/ggmagnify, last access: 17 September 2025; Hugh-Jones, 2023) R packages.

*Author contributions.* KR performed the code and the analysis under the supervision of FGS. KR wrote the first draft and both authors edited and reviewed the final version.

*Competing interests.* The authors declare that no competing interests are present.

*Acknowledgements.* The French Région Hauts-de-France and the ANR project $CO_2$Coast (principal investigator Hubert Loisel, grant no. ANR-20-CE01-0021) are acknowledged for cofunding of the PhD thesis of Kévin Robache. Yann Bozec, Sarah Bureau, and Jean-Philippe Gac are acknowledged for their assistance regarding the technical aspects of the Astan buoy.

*Financial support.* This research has been supported by the Région Hauts-de-France and by the Agence Nationale de la Recherche (grant no. ANR-20-CE01-0021).

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
