# Peer review of "Multiscale statistical analysis of thermal and non-thermal components of seawater $pCO_2$ in the Western English Channel: scaling, time-reversibility, and dependence"

_EGUsphere, 2025_

## Author Comment (AC1)

**Multiscale statistical analysis of thermal and non-thermal components of seawater $p\mathrm{CO_2}$ in the Western English Channel: scaling, time-reversibility, and dependence**

—

**Reply to Referees' Comments**

Kévin Robache and François G. Schmitt

We would first like to thank Referee 1 (R1) and 2 (R2) for their valuable comments and for their engagement in the review process. In the sections below, to clearly distinguish between the referees' remarks and our revisions, the referees' comments are highlighted in blue, while our modifications to the manuscript are indicated in green.

This response letter is structured into 3 parts: Section 1 addresses R1's comments, Section 2 covers R2's comments and the last section is about other changes.

**Contents**

**1 Referee 1 comments (RC1)**

Robache and Schmitt apply statistical approaches, that are more commonly used in other fields, to a high-frequency surface ocean biogeochemical data set collected on the coastal shelf off Brittany, France. These approaches are able to quantify relationships between different measured parameters as supporting evidence for the thermal vs non-thermal (e.g., primary production, river input of DIC) drivers of surface ocean $p\text{CO}_2$ variability. The manuscript is well-written, targeted to an audience with some existing expertise in these statistical approaches. I recommend publication of this manuscript after a few issues are addressed.

Thank you very much for this positive comment and for your review. Please find below our responses to each of the points raised.

**1.1 Major comments**

The data set used in this analysis is 5 years long, with gaps (Table 1). At best, these statistical analyses address sub-annual $p\text{CO}_2$ variability and its potential drivers. This needs to be clearly stated and incorporated into the interpretation of results.

We thank the reviewer for this helpful comment. The following sentence was added in the discussion to clarify this point: "Nevertheless, given the 5-year length of the dataset and the presence of gaps, our analysis is best suited to characterizing sub-annual variability and its drivers. Longer continuous time series would be required to extend these analyses to larger temporal scales for which other dynamics could be expected (e.g., long-term memory at decadal timescale; Séférian et al., 2013)." (lines 374–377).

For example, how does the lack of a longer-term data set impact the finding that some of the time series are irreversible at time scales longer than one month? Could this be influenced by lack of data that would have longer-term signals, like interannual signals? How do the results of this analysis challenge or build upon past studies that have looked at these signals, such as the long-term memory processes that drive ocean biogeochemistry (e.g., the higher-order auto-regressive processes shown by Séférian et al., 2013, doi.org/10.5194/esd-4-109-2013, or the diurnal to seasonal processes shown by Torres et al., 2021, doi.org/10.1029/2020GL090228)?

Time reversibility and long-term memory are two distinct properties of time series that are not mutually exclusive. A process may display long-term memory while lacking time symmetry.

Conversely, for a Gaussian process such as fractional Brownian motion, long-term memory can coexist with a reversible structure.

However, the following sentence has been added to the discussion to address this point: "Longer continuous time series would be required to extend these analyses to larger temporal scales for which other dynamics could be expected (e.g., long-term memory at decadal timescale; Séférian et al., 2013)." (lines 375–377). In addition, the reference Torres et al. (2021) has also been added to the introduction (line 39), as it was relevant.

This manuscript is written for a reader with some knowledge of advanced statistical approaches. In order to reach a wider audience of ocean biogeochemists, it would be useful to ensure all statistical and mathematical jargon is defined, such as conditional means (section 3.2.1) and power-law slopes and passive scalar turbulence (section 3.1.3).

Some sections have been added to better define the mathematical concepts used. First, a paragraph was added to define the conditional mean: "Conditional means are valuable exploratory tools. They provide a way to estimate the overall, often nonlinear, dynamics of a variable $u$ with respect to another variable $v$ (Wand and Jones, 1994):

$$m_v(x) = \langle v \,|\, u = x \rangle, \tag{1}$$

For empirical time series, an interval of values $(a, b)$ can be used instead of a fixed value $x$ in order to estimate the conditional mean:

$$m_v(x) = \langle v \,|\, u \in [a, b] \rangle, \;\; \text{with } x = \tfrac{1}{2}(a + b) \tag{2}$$

This approach allows one to capture trends in the conditional dynamics between two variables which can be nonlinear, and cannot, by definition, be captured by a linear regression. The conditional average is usually computed at a given scale (here, over $[a, b]$), so that smaller-scale fluctuations are filtered." (lines 145–154).

Definitions of a passive scalar ("i.e., a scalar that is advected by the flow but does not influence its dynamics", line 128), of a power-law ("i.e., $y = x^{\beta}$", line 206), and of a lemniscate ("i.e., a curve shaped like a infinity symbol", line 283) were also provided.

The authors note the importance of large observation databases such as SOCAT (line 30).

One benefit of a quality-controlled, internally-consistent data synthesis effort such as SOCAT is data accessibility for regional comparisons. The impact of this work would be enhanced by submitting the ASTAN buoy data to SOCAT...

Thank you for this comment. Unfortunately, we do not own the data. We have therefore forwarded this suggestion to the concerned colleague. Nevertheless, the data are freely available from the SEANOE repository (Bozec et al., 2025).

...and exploring $pCO_2$ variability and drivers of similar time series found in SOCAT, especially high-frequency time series like from the Thornton buoy 10 km off Belgium coast that measures the same parameters and may have similar local drivers such as nearby freshwater sources.

The Astan buoy was chosen for our study because it provides simultaneous high-frequency measurements of $pCO_2$ and temperature over a sufficiently long period of time. This study also serves as a methodological example for our various analyses. A comparison with other buoys would of course be interesting, but we see this more as a perspective for our work, allowing us, for example, to compare results between basins (the Thornton buoy is located 500 km from the Astan buoy, in another basin, located within an offshore wind farm 30 km from the coast; `https://deims.org/177ff4a8-9481-495e-a55a-ec7d32bf6e30`, last access: 27 August 2025).

**1.2 Minor comments**

Lines 65-72: States what depths and heights these measurements are collected[1]. Also describe the data quality assurance and control process[2]. Were outliers identified and flagged?[3] With a fluorometer deployed at the surface, how is interference from ambient light addressed?[4]

1. SST, SSS, oxygen, and fluorescence measurements were carried out at a depth of 5 m, while $pCO_2$ measurements were taken at 6 m. The atmospheric pressure was measured at 15 m height. These details have been incorporated into the text (lines 67, 69, 72 and 75).

2. Details of the calibration of chl $a$ measurements from fluorescence data have also been included: "based on fluorescence and calibrated using low-frequency chlorophyll data; see Gac et al. (2020)." (lines 70–71). Also, a section was added on the technical details of

data acquisition by Gac et al. (2020): "Moreover, blank measurements were regularly performed in distilled water for quality control, with sensors retrieved every three months for inspection, cleaning, battery checks, and reagent level verification. More technical details can be found in Gac et al. (2020)" (lines 77–79).

3. Outliers (i.e., aberrant values due to sensor errors) were removed from the final dataset and were not considered in the analysis.

4. A section was added to the text to clarify this point: "To mitigate potential issues related to ambient light, a cover was placed over the fluorescence sensor and the measurements were carried out in an opaque chamber." (lines 72–73).

Line 184: Explain why the daily cycle influence is avoided in the Fourier analysis but included elsewhere (e.g., tidal signal in lines 193-195 and in the reversibility section)?

Fourier spectral analysis allows for the precise isolation of dynamics associated with different scales. In Fourier space, the daily cycle appears as a distinct peak in the frequency domain. Thus, the periodicity is not eliminated by the analysis itself. However, such peaks can interfere with the regression-based estimation of the power-law slope. To address this, we restricted the regressions to frequency ranges that exclude the intervals where these peaks occur.

In contrast, for the other analyses (e.g., irreversibility curves; see Figs. 7 and 8), the periodicity is not localized in time and therefore cannot be selectively identified or excluded. In this case, however, the objective is not to estimate scaling laws but rather to characterize the overall empirical dynamics of the indicator. Consequently, the influence of the tidal cycle is less disruptive for the analysis.

Line 192: Clarify what is meant by "small-scale" here.

We have added "(e.g., from hours to months; Robache et al., 2025)" (line 217).

Line 229: Remove "and thus the oceanic biological pump." The relationship between $pCO_2$ and chl $a$ only suggest the effects of primary production in surface waters. The biological pump involves several other processes connecting those surface processes to sinking organic carbon, remineralization, and eventual sequestration at depth. The data presented here do not capture those subsurface processes.

Thanks for this valuable comment. This has been removed as requested.

Done.

Done: "and is published in Bozec et al. (2025) and Robache and Schmitt (2025)" (lines 413–414).

**2    Referee 2 comments (RC2)**

The study 'Multiscale statistical analysis of thermal and non-thermal components of seawater $pCO_2$ in the Western English Channel: scaling, time-reversibility, and dependence' statistically analyzes the $pCO_2$ data from the ASTAN cardinal buoy (Brittany, west coast of France) with at 30-minute intervals. The total data is available for 5 years. The author applies various statistical approaches to quantify the relationship between the ocean state variables influencing the thermal and non-thermal components of the $pCO_2$ variability. The paper is overall well written but is statistically too heavy. But depreciating the scientific explanations or essence for performing these statistical analyses. In general, any statistical analysis is performed to explain an underlying problem. This study lacks these explanations and seems like a statistical methodology that was simply applied to a set of variables, for the sake of applying. However, I believe the manuscript could be an important contribution if the authors are able to enhance the explanation highlighting the need for this hefty statistical analysis to understand the physics-based connection between different variables. My major and minor comments are as follows.

We would first like to thank the reviewer for this feedback on our study, and for these positive comments. We have made changes to better explain the different analyses that have been implemented and also the transitions between them. The changes introduced are listed below.

- "with the aim of characterizing the dynamics of the time series of the different scalars across temporal scales" (lines 204–205)

- "Based on the results reported from the Fourier spectral analysis, the dynamics of each of the scalars considered in this study is considered using tools from the field of turbulence. Recently, the property of irreversibility—essential for characterizing the temporal symmetry of signal statistics—has been investigated in relation to turbulence (Schmitt, 2023).

Indeed, in nature many phenomena can be modeled by a nonlinear relaxation dynamics, characterized by a rapid growth phase followed by a slower decay. Examples include epidemic outbreaks (Bestehorn et al., 2022), river discharge (Mathai and Mujumdar, 2022), or El Niño–Southern Oscillation (ENSO; An and Jin, 2004) variability in climatology. The characterization of time-reversal asymmetry in oceanography is thus of relevance in this context. To our knowledge, this type of analysis has never been applied to *in situ* environmental time series, and the turbulent context here offers a valuable opportunity to explore this property for the series under consideration." (lines 232–240)

- "In the previous sections, we focused on the univariate statistical properties of the scalars. In this section, we turn to the statistical relationships that may exist between different scalars, using a range of methods, some more common than others, with the aim of characterizing them." (lines 261–263).

Please find below our responses to your major and minor comments.

**2.1 Major comments**

Although the authors claim to have analyzed 5 years of data, Table 1 clearly shows that the primary variable, i.e., $p\mathrm{CO}_2$ of this study, have $> 60\%$ gap. This raises a serious question on the significance of the statistics performed in this study. The authors should explain these limitations and how does it impact the overall outcome of this study.

All the statistical analyses employed are suitable for datasets containing missing values. Most of these methods rely on increments, meaning that the missing data are not directly incorporated. Moreover, because the dataset is relatively large and we are not estimating inferred statistics over very long temporal scales (e.g., greater than one year), the results remain robust, with averages computed over large numbers of observations.

In addition, the data gaps are irregular—some short, others long—and this pattern is consistent across all temporal scales considered. Consequently, all increments are statistically represented in our results. The following text has been added: "In addition, the intervals between missing values vary considerably in length, with some being long and others short. As a result, the influence of missing values is distributed across all scales. Overall, despite approximately 60 % of the $p\mathrm{CO}_2$ data being missing, the statistics at each scale (computed from the increments between available values) are still based on a sufficiently large number of observations, ensuring

statistical convergence." (lines 86–89).

Table 1 also highlights that the data gap varies for different variables. The authors perform PDF quotient which is based on joint probability. The variable data availability across the different variables may have a significant impact on such statistical analysis. The authors should make equal data length and then perform such PDF quotient. This may or may not change the outcome of this study, but such analysis is quint essential to show. Also, please provide figures showing the available data count for different time periods. This will give an idea how the distribution of data may impact the seasonality or the interannual or higher order variabilities. We thank the reviewer for this valuable comment. We carried out the requested analysis and found that, in our case, the results remained goobally unchanged. In fact, some information is even lost when relying solely on simultaneous observations of the different scalars. We therefore retained the results of our original analyses in the main text, but added the new figures in an appendix. Regarding the temporal aspect, the purpose of using probability distribution–based methods here is to eliminate dependence on time. The series is considered as a whole, and the results are estimated in the $(x, y)$ plane, without taking time into account. However we agree that if there are missing data in some specific season this may have impacts on the statistics (some values could not be represented). Some text has been added to the new Appendix A section. The following text has been added, containing the details of this new analysis and the comparison with the previous results:

"In the analysis presented in Sect. 3.2.2, the numbers of value pairs used to estimate the PDF quotients are not all equal, due to the fact that the missing data are not the same for all series. To check the robustness of our results, we therefore compared these results with those obtained by considering only the observations for which all parameters were simultaneously present (no missing data in any of the parameters). First, a comparison of the number of pairs used for each parameter in both cases is presented in Table 1 (Table A1 in the revised manuscript). In the case where only simultaneous observations of all scalars are retained (case B in the table), the number of available observations decreases due to missing data for chl $a$ and $O_{sat}$. These missing values do not occur at the same periods, which explains this reduction. We examined the monthly distribution of the available data pairs in each case. The results are shown in Fig. A (Fig. A1 in the revised manuscript) . Panel (a) corresponds to case A. For this case, the distribution of data pairs of SST, chl $a$, and $O_{sat}$ with $X_{sat}$ is provided. It is not shown

for the other scalar variables since their distributions are similar to that of SST. Notably, in case A, most months are covered for all scalars except for $O_{sat}$, for which December is missing. However, the temporal coverage is not completely uniform at the monthly scale. It is somewhat more consistent at the seasonal scale. For instance, during winter, the values from January and February allow winter conditions to be represented in the results. For case B, shown in panel (b), the distributions are overall similar to those obtained previously for chl $a$ and $O_{sat}$, except for November and December, when very few values are retained due to the homogenization of the time series across the different scalars (as explained earlier).

**Table 1:** Number of pair values used to estimate the PDF quotient (compared with $X_{sat}$ values) for each scalar, either by retaining all available pairs (case A) or by considering only the values for which simultaneous measurements of all scalars were available (case B).

| Scalar | Case A | Case B |
|---|---|---|
| SST | 32 582 | 18 292 |
| SSS | 32 582 | 18 292 |
| $P_{atm}$ | 32 582 | 18 292 |
| $p CO_2$ | 32 582 | 18 292 |
| Chl $a$ | 23 238 | 18 292 |
| $O_{sat}$ | 26 480 | 18 292 |

Following this, we repeated the PDF quotient analysis, this time considering only the values recorded simultaneously. The results obtained for SSS and chl $a$ are presented in Fig. B (Fig. A2 in the revised manuscript). Overall, the results from this analysis remain consistent with those obtained in Sect. 3.2.2. A relationship is still observed between high $X_{sat}$ values and SSS, as well as between high chl $a$ values and low $X_{sat}$ values. The results for the other scalars are not shown here for readability, but the conclusions remain the same for each of them.

Finally, by relying on a smaller number of observations, some important information is actually lost regarding the interaction between SSS and $X_{sat}$ (and similarly for atmospheric pressure and SST), for which more values are available (i.e., all $X_{sat}$ values). For this reason, in this study we retained in the main text the analyses that included the largest number of values."
(Appendix A, lines 420–443).

Moreover, we added this sentence in the discussion section: "It is also necessary for these time series to be as temporally homogeneous as possible (i.e., with no or few interruptions) in order to enhance the robustness of the applied statistical analyses. Indeed, in the present case, the data used in the PDF quotient analysis are, for example, not perfectly homogeneous at the

[Figure]

**Figure A:** Percentage of observations (%) per month of the year, considering (a) all available $X_{\text{sat}}$ data (case A) and (b) only the subset of data with complete observations for all scalars (case B).

monthly scale (see Appendix A) due to interruptions in the time series, themselves resulting from the difficulties of sampling in the marine environment." (lines 377–381).

Please explain the meaning of the time-reversal symmetry and asymmetry. Also, explain why such analysis is important. What scientific problem is addressed or highlighted, with such analysis? Especially keeping in mind variables such as $p\text{CO}_2$. The return period analysis is generally performed for storms, cyclones, sea level rise, etc. These suggest the time required for reoccurrence of such events that may impact the coastal population/society.

Irreversibility is an important property in signal analysis, as it allows characterization with respect to the arrow of time. This property can also have implications for modeling: if a signal is irreversible, its statistics differ depending on the direction of time. This is the case, for example, for "ramp–cliff" structures in passive scalar turbulence, which are characterized by a phase of slow variations of a scalar followed by an abrupt change (see for example Kang et al., 2014). Many natural phenomena can be described by nonlinear relaxation dynamics, character-

[Figure]

**Figure B:** Representation of the PDF quotient $Q(x, y)$ estimated for $X_{sat}$ and (a) SSS and (b) chlorophyll $a$, using only the data for which all scalars were measured simultaneously (case B). The horizontal black dashed lines represent the 100 % value. For each variable, the chosen bandwidth used for the calculation of the 2D-PDF is defined as 1 % of the difference between its maximum and minimum values. The x-value and y-value are the mean of each bounds of each range. A contour plot of the value $|Q| = 0.3$ is represented in each panel and the value below this threshold are represented in gray, to separate visually the area where $10^{|Q|} < 2$.

ized by a rapid growth phase followed by a slower decay. Examples include epidemic outbreaks (Bestehorn et al., 2022), river discharge (Mathai and Mujumdar, 2022), or El Niño–Southern Oscillation (ENSO; An and Jin, 2004) variability in climatology. These structures are clearly irreversible (they are not symmetric with respect to time), and modeling them forward or backward therefore raises different issues. In our case, this analysis also highlighted results that we consider important, since biological and/or chemical processes appear to be more reversible than physical processes. The following text has been added: "essential for characterizing the temporal symmetry of signal statistics" (lines 233–234) and "Indeed, in nature many phenomena can be modelled by a nonlinear relaxation dynamics, characterized by a rapid growth phase followed by a slower decay. Examples include epidemic outbreaks (Bestehorn et al., 2022), river discharge (Mathai and Mujumdar, 2022), or El Niño–Southern Oscillation (ENSO; An and Jin, 2004) variability in climatology. The characterization of time-reversal asymmetry in oceanography is thus of relevance in this context." (lines 234–238).

Finally, return-time analyses do not incorporate the question of reversibility in time series, as they only focus on one temporal direction.

Between lines 145-150 'This provides information that complements what is obtained from the correlation. While the correlation is a single numerical...' authors present some key conclusions. However, it is important to highlight why is it important to know which values are in $(x, y)$ plane exhibit weak or strong relationship.

The correlation (or covariance) between two series provides only a single number, without accounting for how both variables may be related across their range of variation. Here, the Probability Density Function (PDF) quotient makes it possible to identify the ranges of values over which the scalars are dependent on each other, and the strength of this dependence. This was already stipulated in the dedicated "Data and Methods" section; however, we have now added the following text in the revised manuscript to also explain and emphasize this point in the dedicated "Results" section: "the correlation (or covariance) between two series provides only a single number, without accounting for how both variables may be related across their range of variation. Here, the PDF quotient makes it possible to identify the ranges of values over which the scalars are more or less dependent on each other, and the strength of this dependence." (line 286–288).

Lines 159-160 'The CV of $p\mathrm{CO}_2$...' this could be because of the high missing values in $p\mathrm{CO}_2$.

Even though some data are missing in the series, it still contains 32 582 observations, which makes it valuable for estimating a mean and a standard deviation, allowing the calculation of the coefficient of variation, which provides a normalized measure that can be compared across time series. The number of values used here is in fact larger than the size of most datasets employed in our previous study, with which we make a comparison (see Table A1 in Robache et al., 2025). Therefore, the missing data are not responsible for the "low" coefficient of variation observed here.

**2.2 Minor comments**

Provide full form of ASTAN and WEC on its first occurrence in the manuscript.

This is done for WEC (line 59 instead of 184). There is no acronym for ASTAN, it is only the

name of the buoy. To avoid any ambiguity, we have replaced all occurrences of "ASTAN" with "Astan".

Improve Fig. 2, why is it required?

We thank the reviewer for this comment. Indeed, we had neglected to specify the purpose of this figure. The following sentence has been added to the text: "Figure 2 presents the value of $A$ derived in this study, along with the regression between SST and $\log(p\mathrm{CO}_2)$, compared with *in situ* measurements and the regression reported by Takahashi et al. (1993)." (lines 100–102).

Line 161 '... $X_{\mathrm{sat}}$ time series still exhibits significant fluctuations...' How does the author decide that the fluctuations are significant?

We thank the reviewer for this comment. By "significant" we did not mean a statistical test of significance. Rather, we intended to convey that the $X_{\mathrm{sat}}$ time series still shows a non-negligible dynamical behavior, i.e. the variations are clearly observable and do not vanish. To avoid ambiguity, we have revised the wording in the manuscript: "the $X_{\mathrm{sat}}$ time series still exhibits noticeable fluctuations" (line 182). In the revised manuscript, all other occurrences of "significant" were also replaced or removed (lines 61, 189, 192 and 279).

Elongate Fig. 3 along $x$-axis for clarity of image.

The $x$-axis has been elongated and the size of the figure has been increased to improve its clarity (see Fig. C, corresponding to the new Fig. 3 of the revised manuscript).

Line 171 'In general, this indicates that the non-thermal component...' Explain how does the author reach this conclusion?

We thank the reviewer for this remark. The phrase "In general" was not the most appropriate formulation here. What we meant is that, when considering the full time series, the non-thermal component of $p\mathrm{CO}_2$ still displays noticeable fluctuations. This indicates that it follows its own dynamics rather than being fully suppressed. To avoid ambiguity, we have replaced "In general" with "Overall" (line 192) in the revised manuscript.

Lines 175-177 'On the contrary..' The results reported should be visible from Fig. 3

To improve the readability of the figure, we added purple horizontal lines corresponding to April

[Figure]

**Figure C:** The complete $X_{\text{sat}}$ time series, as defined in Eq. (3), represented as a percentage. The red dashed line represents the 100 % value which is reached when $p\text{CO}_2 = p\text{CO}_2{}^{\text{thermal}}$. The inset is a zoom over a period of 10 days showing that the series presents fluctuations even at small scales. Purple horizontal lines indicate April and October of each year, providing quick visual reference points along the $x$-axis, which is spaced at monthly intervals.

and October (Fig. C), two months frequently mentioned in the text. The $x$-axis ticks are spaced one month apart, providing a quick temporal reference that facilitates visualization of the results discussed. This has been added to the Fig. 3 caption: "Purple horizontal lines indicate April and October of each year, providing quick visual reference points along the $x$-axis, which is spaced at monthly intervals.".

Line 178 should be Figs 4a and 4b

Done: "Figs 4a-b" has been changed to "Figs 4a and 4b" (line 199).

Lines 179-180 'More...' The conclusion presented here is subjected to the amount of data availability and could change if more data is available. Authors should consider this.

We agree with this remark; however, since the number of datapoints is large, we assumed that statistical convergence is reached. Here, the reviewer's comment may have been referring also to the missing values. To address this, we have added the following sentence to the text: "However, this result should be interpreted with caution due to the presence of missing data in the

series." (lines 201–202).

Line 184 says that for scaling data below daily was discarded, more explanation is required for this statement. Especially, when the authors have very high-resolution data.

The $\beta$ value characterizes the scale invariance of each series. We limited the regression range to estimate the scaling slope to scales larger than 1 day to avoid the effect of the daily cycle peak, which can disrupt the scaling. Furthermore, at very high frequencies, the spectral dynamics are not always accurately reproduced due to missing data and the limited number of observations. This has been added in the text: "The highest frequencies (below the daily scale) are not considered to avoid the effects of missing data and the daily and tidal periodicities, which can disrupt the scaling properties (Schmitt and Huang, 2016)." (lines 207–209).

Line 192, please explain 'small-scale intermittency' in detail. What is the physical relevance of the same?

The following section was added in the text: "In the turbulence framework, intermittency is an intrinsic property of turbulent time series (Frisch, 1995). Intermittency in turbulence was first revealed through early experimental measurements (Batchelor and Townsend, 1949): the term refers to strong, localized fluctuations in the velocity field, as well as in turbulent scalars. More broadly, this means that extreme fluctuations can be observed across different scales, occurring much more frequently than would be expected for a Gaussian process." (lines 218–222).

Lines 217-218 'As for the Fourier spectra...' Explain the physical relevance to obtain information at such small scales.

The very fine scales correspond to the temporal scales at which chemical and/or biological reactions can occur, in addition to physical processes. The purpose of conducting analyses at such fine scales is to capture the combined effect of all these mechanisms within the temporal continuum.

Also, since $pCO_2 = f(SST)$ isn't the observation reported in this line but obvious?

The link between the thermal component of the partial pressure of $CO_2$ and SST is, of course, expected given its definition. However, it is still important to highlight this for several reasons. First, because of the implications: the thermal component follows a spectrum identical to that

of SST. Then, any deviation of the $pCO_2$ spectrum from SST indicates that non-thermal processes have a significant influence on $pCO_2$ dynamics.

Second, despite the relationship between the two variables, the results can still differ. For example, in the reversibility analysis, although the dynamics of the indicator $Po$ are similar for both variables, they are not exactly the same, particularly regarding the extreme values reached.

Line 229, please remove the 'biological pump' reference. This data is only at surface, and a detailed analysis of the sub-surface data is required to confirm the role of the 'biological pump'.

Done.

**3    Other changes**

Some mistakes have been corrected in the text:

- The number of atmospheric pressure missing values has been corrected (Table 1 and line 84).

- "log" has been remplaced by "$\log_{10}$" in Eq. (9).

- "0.2" has been remplaced by "0.3" (line 292).

- "Yann Bozec, Sarah Bureau, and Jean-Philippe Gac are acknowledged for their assistance regarding the technical aspects of the Astan buoy." was added in the "Acknowledgements" section (lines 448–449).

- The R code for the PDF quotient analysis has been published here: `https://github.com/KevinRobache/PDF_Quotient_Code` (last access: 11 September 2025) was added in the "Code and data availability" section (lines 415–416).

- The reference "Calvin et al., 2023" has been updated (now "IPCC, 2023", line 27)

**References**

An, S.-I. and Jin, F.-F. (2004). Nonlinearity and Asymmetry of ENSO. *Journal of Climate*, 17(12):2399–2412. doi:10.1175/1520-0442(2004)017<2399:NAAOE>2.0.CO;2.

Batchelor, G. K. and Townsend, A. A. (1949). The nature of turbulent motion at large wavenumbers. *Proceedings of the Royal Society of London. Series A. Mathematical and Physical Sciences*, 199(1057):238–255. doi:10.1098/rspa.1949.0136.

Bestehorn, M., Michelitsch, T. M., Collet, B. A., Riascos, A. P., and Nowakowski, A. F. (2022). Simple model of epidemic dynamics with memory effects. *Physical Review E: Statistical Physics, Plasmas, Fluids, and Related Interdisciplinary Topics*, 105(2):024205. doi:10.1103/PhysRevE.105.024205.

Bozec, Y., Loisel, S., and Bureau, S. (2025). ASTAN cardinal buoy (western English Channel, France) data from 2015 to 2019 [data set]. doi:10.17882/106537.

Frisch, U. (1995). *Turbulence: The Legacy of A. N. Kolmogorov*. Cambridge University Press. ISBN 978-0-521-45713-2.

Gac, J.-P., Marrec, P., Cariou, T., Guillerm, C., Macé, É., Vernet, M., and Bozec, Y. (2020). Cardinal Buoys: An Opportunity for the Study of Air-Sea $CO_2$ Fluxes in Coastal Ecosystems. *Frontiers in Marine Science*, 7. doi:10.3389/fmars.2020.00712.

IPCC (2023). *Climate Change 2023: Synthesis Report. Contribution of Working Groups I, II and III to the Sixth Assessment Report of the Intergovernmental Panel on Climate Change*. IPCC, Geneva, Switzerland.

Kang, Y., Belušić, D., and Smith-Miles, K. (2014). Detecting and Classifying Events in Noisy Time Series. *Journal of the Atmospheric Sciences*, 71(3):1090–1104. doi:10.1175/JAS-D-13-0182.1.

Mathai, J. and Mujumdar, P. P. (2022). Use of streamflow indices to identify the catchment drivers of hydrographs. *Hydrology and Earth System Sciences*, 26(8):2019–2033. doi:10.5194/hess-26-2019-2022.

Robache, K. and Schmitt, F. G. (2025). Thermal and non-thermal components of $pCO_2$ estimated from ASTAN cardinal buoy (western English Channel) data from 2015 to 2019 [data set]. doi:10.17882/106550.

Robache, K., Schmitt, F. G., and Huang, Y. (2025). Scaling and intermittent properties of oceanic and atmospheric $pCO_2$ time series and their difference in a turbulence framework. *Nonlinear Processes in Geophysics*, 32(1):35–49. doi:10.5194/npg-32-35-2025.

Schmitt, F. G. (2023). Scaling Analysis of Time-Reversal Asymmetries in Fully Developed Turbulence. *Fractal and Fractional*, 7(8):630. doi:10.3390/fractalfract7080630.

Schmitt, F. G. and Huang, Y. (2016). *Stochastic Analysis of Scaling Time Series: From Turbulence Theory to Applications*. Cambridge University Press. ISBN 978-1-107-06761-5.

Séférian, R., Bopp, L., Swingedouw, D., and Servonnat, J. (2013). Dynamical and biogeochemical control on the decadal variability of ocean carbon fluxes. *Earth System Dynamics*, 4(1):109–127. doi:10.5194/esd-4-109-2013.

Takahashi, T., Olafsson, J., Goddard, J. G., Chipman, D. W., and Sutherland, S. C. (1993). Seasonal variation of $CO_2$ and nutrients in the high-latitude surface oceans: A comparative study. *Global Biogeochemical Cycles*, 7(4):843–878. doi:10.1029/93GB02263.

Torres, O., Kwiatkowski, L., Sutton, A. J., Dorey, N., and Orr, J. C. (2021). Characterizing Mean and Extreme Diurnal Variability of Ocean $CO_2$ System Variables Across Marine Environments. *Geophysical Research Letters*, 48(5):e2020GL090228. doi:10.1029/2020GL090228.

Wand, M. P. and Jones, M. C. (1994). *Kernel Smoothing*. CRC Press. ISBN 978-0-412-55270-0.

---

## Author Response (AR3)

**Multiscale statistical analysis of thermal and non-thermal components of seawater $p\mathrm{CO_2}$ in the western English Channel: scaling, time-reversibility, and dependence**

—

**Reply to Referees' Comments**

Kévin Robache and François G. Schmitt

We thank the Associate Editor and reviewer for their time and commitment throughout the review process. Please find our responses to the concerns below. The reviewer's comments are shown in blue, and the additions made to the manuscript are highlighted in green.

**Reviewer's comment**

While I appreciate the considerable effort the authors have devoted to revising this manuscript, I find their response to my first question unsatisfactory. The authors claim to have analyzed five years of data; however, Table 1 clearly shows that the primary variable, $p\mathrm{CO_2}$, contains over 60 % missing data. Such a substantial data gap raises serious concerns about the robustness and interpretability of the statistical analyses presented. The authors must more thoroughly address these limitations and assess their potential impact on the study's overall conclusions.

In their reply, the authors argue that the statistical methods applied remain valid even with missing values. However, this response does not sufficiently address the core concern - namely, how much the results might differ if a more complete or longer-term dataset were available. The authors further contend that "because the dataset is relatively large and we are not estimating inferred statistics over very long temporal scales (e.g., greater than one year), the results remain robust, with averages computed over large numbers of observations." This justification is unconvincing and appears overly simplistic. Oceanic state variables such as $p\mathrm{CO_2}$ are strongly influenced by seasonal variability, suggesting that results could change considerably

if data from all months or additional years were included. Moreover, interannual variations - potentially driven by anthropogenic influences - may further affect the observed patterns and trends.

We did not fully understand the reviewer's comments on this subject during the previous round. We hope we now have a better understanding of the reviewer's reservations, and that they will find our revision more satisfactory. In our response here, we separate seasonal and interannual variations.

With regard to interannual variability, we agree with the reviewer that the results obtained could be different if a longer database were available, and that interannual variations, with their potential anthropogenic influences, have an impact on the results. However, this is the case for all studies conducted over a given period: it is clear that we do not have data for another period to know what the interannual influences are. It is therefore clearly not possible for us to quantify this, but we have added comments on this subject in the conclusion, to remind readers that interannual variations, with their potentially different forcings, could have an influence on the reported results.

This has been added in the text: "The statistics reported here may depend on the forcing associated with interannual variability and its potential anthropogenic forcing. Further analyses, performed over different periods of time, will be needed in future works, in order to compare with the results presented here. Also, longer..." (lines 379–381) and "although each time scale is represented by sufficient statistics, the data are not fully homogeneous on a monthly basis. Appendix A shows that December and March were undersampled compared with other months. At the seasonal scale, however, the distributions are more balanced. This temporal heterogeneity represents one limitation of our analysis and highlights the need for future comparisons with datasets containing fewer missing values. Appendix B examines the effects of missing data on the analysis of PDF quotients, restricting the computation to observations for which all parameters were simultaneously available. The influence of missing values is limited in this case, and the results are broadly consistent with those presented in the main analysis." (lines 386–392).

Although the title of the study emphasizes that five years of data were analyzed, the assumption that statistical convergence has been achieved (lines 201-202) is not adequately substantiated. The authors should clarify the criteria used to determine that the available dataset is sufficiently large and representative to support robust statistical inference.

We have added a new Appendix that addresses the issue of missing values and the representativeness of statistics at sub-annual scales. First, despite the fact that there are missing values, some of the methods we use employ increments between two measurement points separated by a given interval. All intervals are thus visited, and statistics are presented for all intervals. The attached figure, which we have added to this Appendix, shows the number of time increments at scale $\tau$ for all scales between 1 day and 365 days. We can see that there are between about 4,000 and 30,000 values considered depending on the scale. This allows us to have relatively good confidence in the convergence of the statistics for all these scales. This has been adjusted in the text: "Further discussion of this issue, including details on the number of data points at each scale, is provided in Appendix A." (lines 91–92).

However, we agree with the reviewer on the potential importance of seasonal variability, and we understand that it is also necessary to verify that each season is well represented in the sampling, so as to avoid giving more statistical importance to one season over another. Figure B below shows the number of values present in the analyzed time series, by month of the year and by season. We can see that certain months, such as March, November, and December, are less frequently present than October or September. This difference becomes less pronounced when we consider seasonal granularity, where the four seasons are relatively more balanced. This is the potentially most significant impact of these missing values, and we have mentioned it both in the Appendix on missing values and in the conclusion.

Here is the new Appendix section: "Most of the analysis methods employed in this study account for the issue of missing values, a common problem when working with high-frequency in situ observations. For instance, spectral analysis, time-reversal symmetry analysis, and the PDF quotient of increments all incorporate the temporal variability of $p\mathrm{CO_2}$ by considering differences between observations separated by a given time scale $\tau$. This approach ensures that, despite occasional data gaps—sometimes extending over relatively long periods—a sufficient number of samples remains available at each time scale. Figure A1 *(Fig. A in this letter)* illustrates the number of $p\mathrm{CO_2}$ increments available for all considered scales between 30 minutes and 365 days. Despite approximately 60 % missing data in the time series (the highest rate among our variables; Table 1), more than 3900 temporal increments are available for timescales shorter than one year. This provides sufficient statistical support for our analyses, which rely predominantly on mean values and are restricted to these timescales.

On the other hand, the distribution of the available $p\mathrm{CO_2}$ data over the five-year period,

shown as a function of month and season in Fig. A2 *(Fig. B in this letter)*, is not perfectly homogeneous. Some months, such as December and March, are under-sampled relative to others, which likely results in an under-representation of their dynamics in the aggregated statistics. At the seasonal scale, the distribution is more balanced, although autumn contributes proportionally more data than the other seasons. This sampling structure should be considered when interpreting the results. A full assessment of its influence would require a continuous time series covering the entire period." (Appendix A; lines 425–439).

[Figure]

**Figure A:** Number of observed temporal increments of $p\mathrm{CO}_2$ as a function of the considered timescale $\tau$.

[Figure]

**Figure B:** Distribution of the available $p\mathrm{CO_2}$ data by month and season. The red dashed line represents the expected values for a uniform sampling.